palaeontology/evolution/taxonomy and systematics

Goniopholididae, aquatic adaptation, hyoid apparatus, secondary palate, gular valve, Neosuchia

**Author for correspondence:**
Junki Yoshida
e-mail: junkiyoshida9@gmail.com

# A new goniopholidid from the Upper Jurassic Morrison Formation, USA: novel insight into aquatic adaptation toward modern crocodylians

Junki Yoshida[1,2], Atsushi Hori[1], Yoshitsugu Kobayashi[1], Michael J. Ryan[3], Yuji Takakuwa[4], Yoshikazu Hasegawa[4]

[1]Hokkaido University, Sapporo, Hokkaido 060-0810, Japan
[2]Fukushima Museum, Aizu-wakamatsu, Fukushima 965-0807, Japan
[3]Department of Earth Sciences, Carleton University, Ottawa, Ontario, Canada K1S 5B6
[4]Gunma Museum of Natural History, Tomioka, Gunma Prefecture 370-2345, Japan

 JY, 0000-0001-5260-0532

Goniopholididae is a group of basal neosuchian crocodyliforms closely related to Paralligatoridae and Eusuchia that lived during the Jurassic and Early Cretaceous. Goniopholidids have the long, flat snout and secondary palate of modern crocodylians, the acquisition of which is regarded as a key feature in the early evolution of crocodylian body plan and their aquatic adaptation. Here, we report a new species, *Amphicotylus milesi*, with the description from the best-preserved specimen to date of Goniopholididae from Wyoming, USA. Its posterior extension of the nasopharyngeal passage (pterygoid secondary palate) and the shortening and dorsal deflection of the ceratobranchial suggest that basal neosuchians could raise their gular valve to separate oral and pharyngeal cavities as in modern crocodylians. The anatomy of *Amphicotylus milesi* sheds light on the acquisition of this new respiratory system in the crocodyliform evolution and their early aquatic adaptation, leading to modern crocodylians.

## 1. Introduction

Modern crocodylians are the descendants of a terrestrial crocodyliform ancestor that gave rise to a lineage of successful semi-aquatic predators through the acquisition of a suite of unique characteristics including a long, flat snout, dorsally facing orbits, closable nostrils, a secondary palate and a gular valve [1]. In modern crocodylians, the pterygoid secondary

palate separates the narial passage from the oral cavity. These regions can be sealed off from each other by raising the ventral fold of the gular valve at the base of the tongue to contact the dorsal fold at the posterior end of the palate. While opening the mouth underwater, the combination of the secondary palate and the closed ventral valve at the base of the tongue allows crocodylians to breathe underwater as long as their external nostrils are above the water surface or holding prey in the mouth by closing the nostril [2,3]. In the evolutionary history of crocodyliforms, the pterygoid secondary palate is present in neosuchian crocodyliforms, such as eusuchians, paralligatorids and derived goniopholidids [4,5]. The evolutionary history of the gular valve is, on the other hand, unknown. The ventral fold of the gular valve embeds a cartilaginous basihyal, which comprises the hyoid apparatus with a pair of ceratobranchials [6]. Although the basihyal does not ossify, ceratobranchials do, but it is poorly known in the crocodyliform fossil record [7]. Without knowledge of the function and evolution of the ceratobranchials, the evolution of the gular valve cannot be determined.

Goniopholididae is a group of basal neosuchian crocodyliforms found in the northern hemisphere during the Jurassic and Early Cretaceous, and a close relative of Paralligatoridae and Eusuchia [8–10]. This family is the earliest crocodyliform group to acquire the modern crocodylian body plan and an inferred semi-aquatic lifestyle [8,11]; therefore, understanding the anatomy, phylogeny and functional morphology of this group are important for revealing the origin of modern crocodylian body plan and lifestyle [12,13].

Here, we describe one of the most complete specimens of Goniopholididae from Wyoming, USA, conduct phylogenetic analysis and scrutinize its internal choana and ceratobranchials, aiming to understand the origin of the gular valve.

# 2. Material and methods

The new goniopholidid specimen (GMNH-PV0000229) was discovered in 1993 from the Upper Jurassic Morrison Formation in the East *Camarasaurus* Quarry, Wyoming, USA [14]. Our data matrix for a phylogenetic analysis was composed of 486 characters and 104 taxa, including GMNH-PV0000229, based on the data matrix by Andrade *et al*. [8]. A heuristic search was performed to obtain the most parsimonious trees using PAUP v. 4.0a [15]. *Gracilisuchus* was used as an outgroup and 24 characters were treated as ordered (see electronic supplementary material). Palatal and hyoid morphologies were quantitatively examined in 87 crocodyliforms (figures 3 and 6; electronic supplementary material). The relative position of the anterior and posterior margins of the internal choana was quantitatively evaluated with standardization by skull width. Hyoid curvature was recorded as a ratio of a length along with the dorsal curved margin of ceratobranchial to the shortest distance between the anterior and posterior ends. Statistical analysis and simulation of trait evolution were performed in R with packages Paleotree [16], Phytools [17] and Strap [18]. Hyoid musculature anatomy and osteological correlates in GMNH-PV0000229 were assessed by dissection of two modern crocodylians (*Crocodylus siamensis* and *Alligator sinensis*).

Institutional abbreviation: GMNH, Gunma Museum of Natural History, Tomioka, Gunma, Japan

# 3. Results

## 3.1. Systematic palaeontology

Crocodylomorpha Walker, 1970 [19]
Neosuchia Benton and Clark, 1988 [20]
Goniopholididae Cope, 1875 [21]
*Amphicotylus* Cope, 1878 [22]
*Amphicotylus milesi* sp. nov.
urn:lsid:zoobank.org:pub:3191C7AA-9786-489E-AFB9-EF1D6F4DD002
Etymology. *milesi* is in honour of Clifford Miles for his contribution to the excavation and museum curating.

Holotype. GMNH-PV0000229, a well-preserved, nearly complete skeleton, missing only coronoids, left palpebral, proatlas, and some caudal vertebrae, left radiale, and some pedal phalanges.

Locality and horizon. Sandstone of the Brushy Basin Member of the Morrison Formation (Kimmeridgian) at the East *Camarasaurus* Quarry, Albany County, Wyoming, USA, yielding a skeleton of *Camarasaurus* (GMNH-PV101) [14].

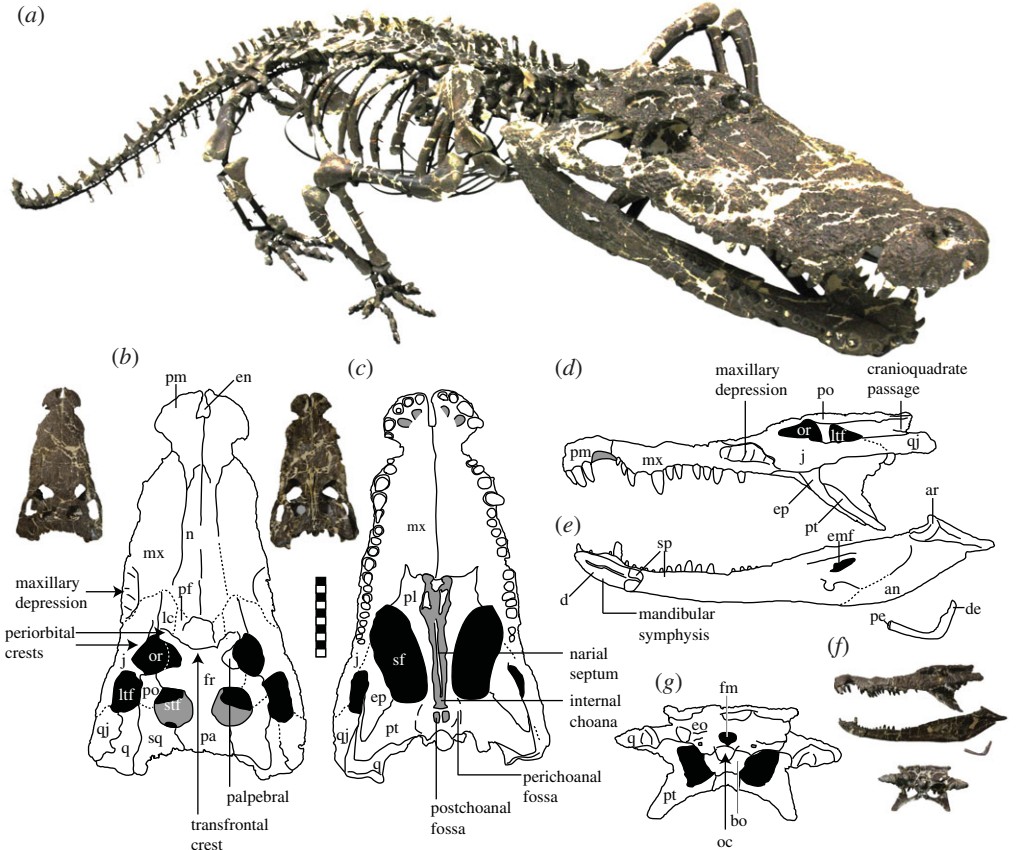

**Figure 1.** A mounted skeleton (*a*), skull in dorsal (*b*), ventral (*c*), left lateral (*d*), mandible in left medial (*e*), ceratobranchial in left lateral (*f*) and skull of *Amphicotylus milesi* (GMNH-PV0000229) in posterior view (*g*). Abbreviations: ar, articular; bo, basioccipital; d, dentary; de, distal end; emf, external mandibular fenestra; en, external naris; eo, exoccipital; ep, epipterygoid; fm, foramen magnum; fr, frontal; itf, infratemporal fenestra; j, jugal; lc, lacrimal; mx, maxilla; n, nasal; oc, occipital condyle; or, orbit; pa, parietal; pe, proximal end; pf, prefrontal; pl, palatine; pm, premaxilla; po, postorbital; pt, pterygoid; q, quadrate; qj, quadratojugal; sf, suborbital fenestra; sp, splenial; sq, squamosal; stf, supratemporal fenestra. Scar bar = 10 cm.

Diagnosis. *Amphicotylus miles* has a unique combination of the following features; broad rostrum, five subfossae in maxillary depression; anterior projection at posterior margin of naris, triangular palpebral, lack of crest C in quadrate, postchoanal fossa, perichoanal fossa being wider than nasopharyngeal passage, and posterior border of internal choana at mid-pterygoid.

## 3.2. Description

The holotype of *Amphicotylus milesi* (figures 1, 2, 4, 5, 7–12) is the most complete skeleton known for goniopholidids. The 43 cm long skull is one of the largest recorded for a goniopholidid. Rostral skull length (preorbital length) is 26 cm (60% of the skull length), indicating medium long rostrum (*sensu* Busbey [23]). The entire dorsal surface of the skull is densely ornamented with circular pits (figure 1*a*). Most sutures of the skull and caudal vertebrae are fused, but the cervical, dorsal and sacral vertebrae remain unfused, indicating immaturity of the individual at death [24,25]. The rostrum curves ventrally as for *Amphicotylus lucasii* [26], and the ratio of rostral width at the middle of the maxilla to the skull width is 0.64, which indicates a broader rostrum than in *Amphicotylus stovalli* (0.5) [27].

As a result of only the right palpebral being preserved, the orbits appear asymmetrical (figure 1*b*). The genuine orbit outline on the right is sub-circular in dorsal view, differing from the elliptical shape of *Goniopholis simus* [28]. The palpebral is the smallest of the bony skull elements and approximately one-third of the orbit in both width and length. The palpebral is triangular in dorsal view (figure 1*b*), as in *G. simus*, *Nannosuchus*, *Anteophthalmosuchus hooleyi* and *Hulkepholis willetti* [28]. The palpebral has a similar morphology to that of *Anteophthalmosuchus hooleyi* in having a concave lateral margin, small size relative to the orbit, and no contact with the lacrimal, but different in the absence of the

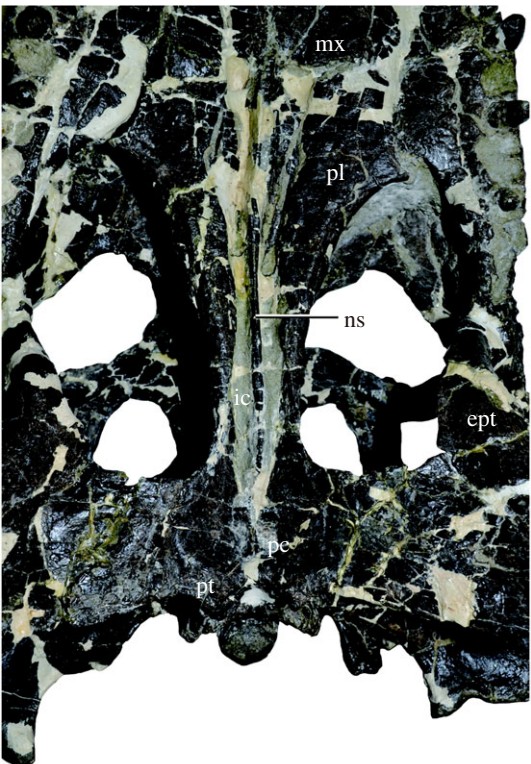

**Figure 2.** Palatine–pterygoid region of *Amphicotylus milesi* (GMNH-PV0000229) in ventral view. Abbreviations: ept, epipterygoid; ic, internal choana; pc, postchoanal fossa; pl, palatine; pt, pterygoid; mx, maxilla; ns, narial septum.

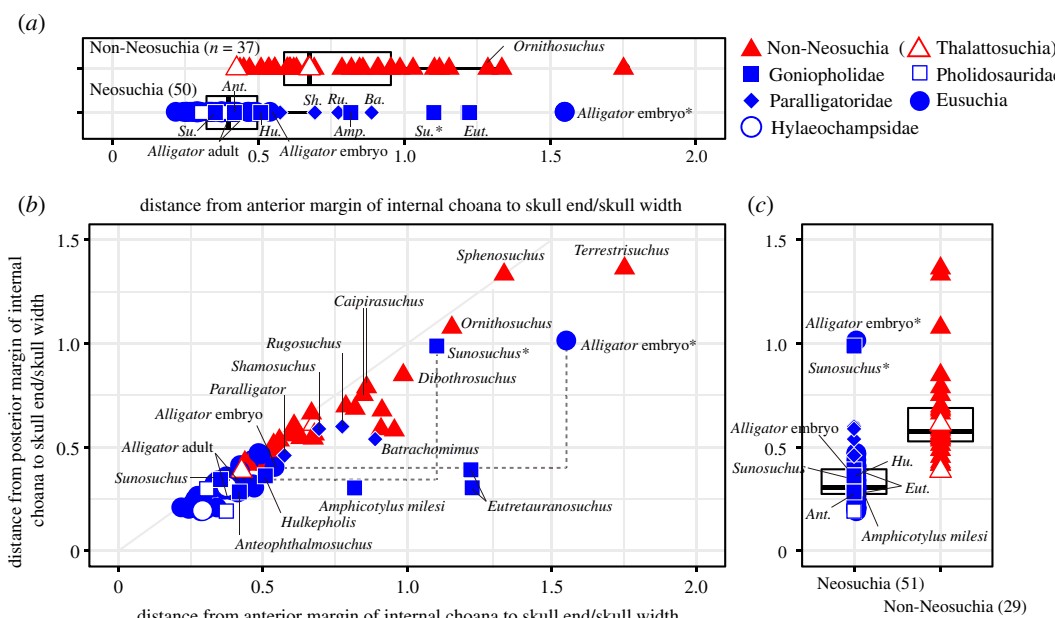

**Figure 3.** Relative positions of anterior and posterior margins of internal choana in non-neosuchians (red) and neosuchians (blue) (*a–c*). Boxplots show median (dark line) and hinges (box range) of relative positions of posterior (*a*) and anterior margins of internal choana (*c*) and a biplot shows relationships between these two marginal positions (*b*). The two groups show significant difference in Wilcoxon rank sum test ($p < 0.0001$) in (*c*). Parenthesis indicates the number of samples. * = anterior foramen. Abbreviations: *Amp, Amphicotylus*; *Ant, Anteophthalmosuchus*; *Ba, Batrachomimus*; *Eut, Eutretauranosuchus*; *Hu, Hulkepholis*; *Ru, Rugosuchus*; *Sh, Shamosuchus*; and *Su, Sunosuchus*.

prefrontal-palpebral contact [28]. The palpebral prevents the frontal from participating in the medial margin of the orbit. The orbits open dorsally and are as large as the supratemporal fossae and the infratemporal fenestrae. In dorsal view, the transfrontal crest of the frontal is continuous with the

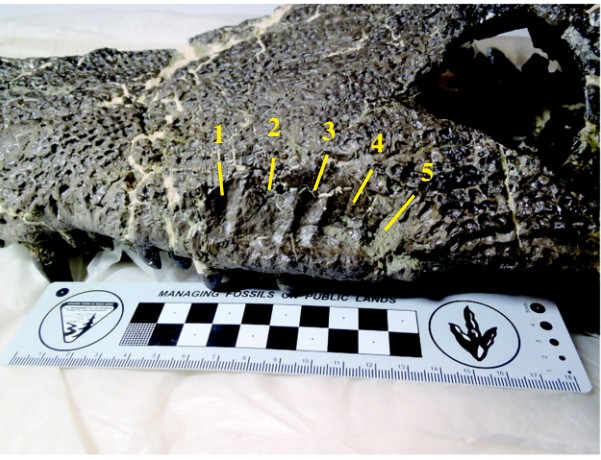

**Figure 4.** Five subfossae of maxillary depression of *Amphicotylus milesi* (GMNH-PV0000229) in left lateral view.

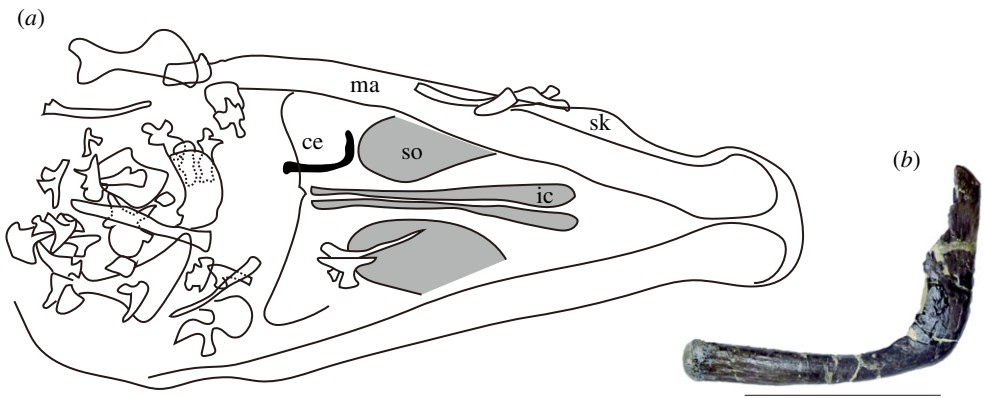

**Figure 5.** (*a*) The discovered location of the ceratobranchial among the cranial bones (GMNH-PV0000229) in ventral view. (*b*) left ceratobranchial in left lateral view. Scale bar, 5 cm. Abbreviations: ce, ceratobranchial; so, suborbital fenestra; ic, internal choana; ma, mandible; sk, skull.

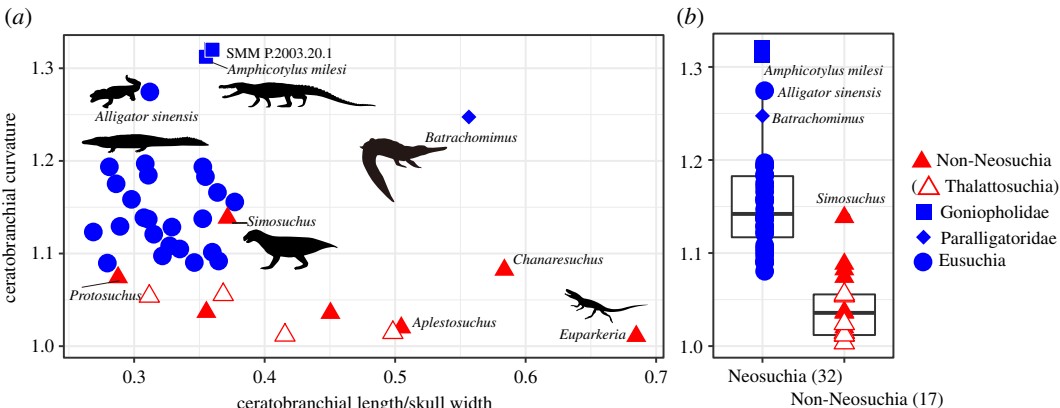

**Figure 6.** Relationship between curvature and relative length of ceratobranchial in crocodyliforms and *Euparkeria* (*a*). Comparison of curvature between non-neosuchians (red) and neosuchians (blue), including *Amphicotylus milesi* (*b*). The comparisons between the two groups show significant difference in Wilcoxon rank sum test ($p < 0.0001$) in (*b*). Parenthesis indicates the number of samples.

upper periorbital crest of the prefrontal and lacrimal along the anteromedial margin of the orbit (figure 1*b*). The crest demarks a stepped change in depth between the anterior process and the main body of the frontal, similar to that seen in *G. simus* and *G. kiplingi* [8]. The supratemporal fenestra is subpolygonal and as wide as it is long. Within the supratemporal fossa, the anterior opening of the temporo-orbital (post-temporal fossa) is visible in dorsal view (figure 1*b*). The infratemporal fenestra

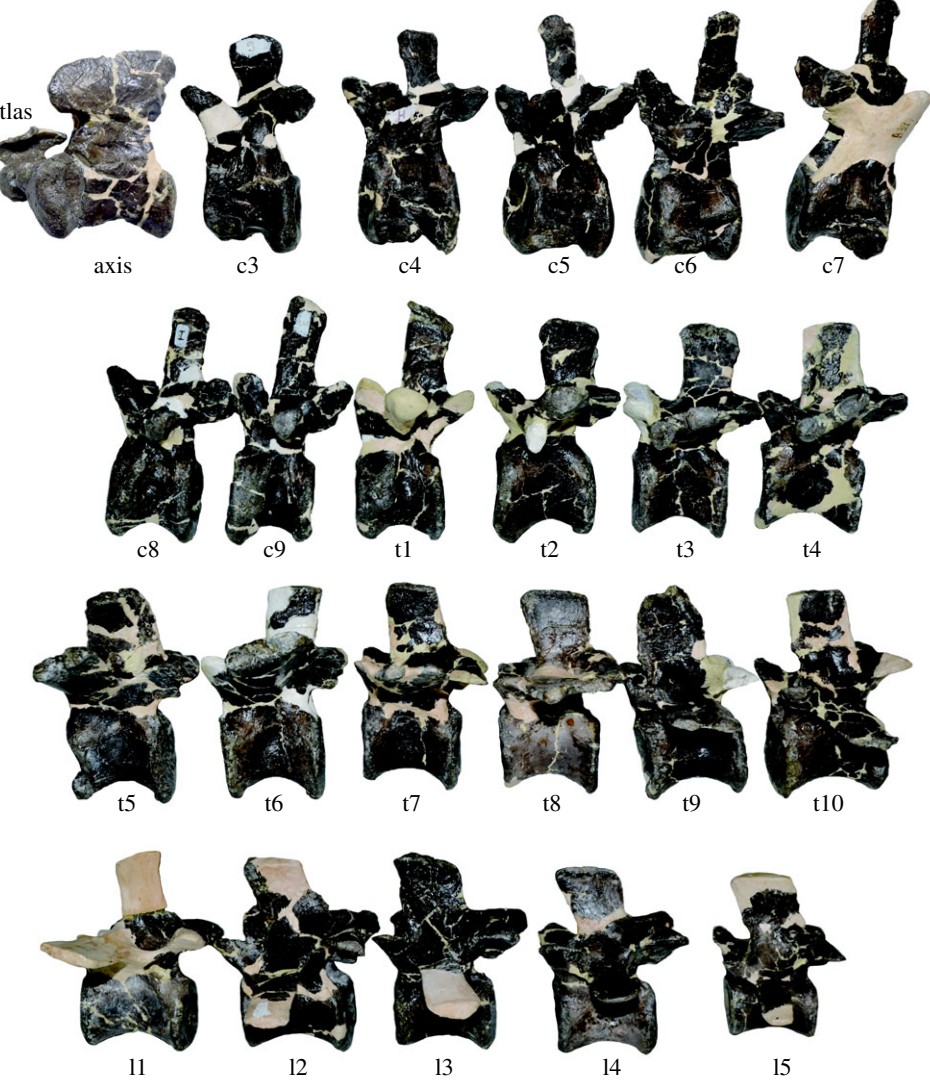

**Figure 7.** Presacral vertebrae of *Amphicotylus milesi* (GMNH-PV0000229) in left lateral views. Abbreviations: c, cervical vertebra; t, thoracic vertebra; l, lumbar vertebra. Scale bar is 10 cm.

is triangular in dorsolateral view. The caudal end of the paroccipital process has the cranioquadrate passage, which opens laterally and is floored by the medial portion of the quadrate, as in goniopholidids and the paralligatorid *Shamosuchus* [29]. The posterior ends of the palatal processes of the maxillae fail to meet in the mid-line (figures 1*c* and 2), as they do in *Calsoyasuchus* [30], *Eutretauranosuchus* [13,31], *Sunosuchus junggarensis* [32] and *Amphicotylus lucasii* [26]. The ventral floor of the nasopharyngeal passage opens at the posterior end of maxilla, palate and pterygoid. The hourglass shape of the opening is also seen in other Jurassic goniopholidids such as *Calsoyasuchus*, *Eutretauranosuchus*, *Amphicotylus lucasii* and *Amphicotylus stovalli* [13,26,30,31]. The ratio of distance between anterior margin of internal choana and posterior skull end to skull width is 0.82, while that of the distance between the posterior margin of internal choana and the posterior skull end to skull width is 0.3 (figure 3). The choanal septum terminates 1 cm before the end of the nasopharyngeal passage in pterygoid (figure 2) and is ventrally smooth as in basal goniopholidids such as *Calsoyasuchus* [30] and *Sunosuchus junggarensis* [32], and in contrast with the bilateral septal laminae in *Eutretauranosuchus* [13] and *Amphicotylus lucasii* [26]. Posterior to the nasopharyngeal passage, the unpaired pterygoid has a small, shallow, rectangular-shaped postchoanal fossa that is not perforated or connected to the passage and sinus.

In common with *Amphicotylus lucasii* and *Amphicotylus stovalli*, the premaxilla of *Amphicotylus milesi* is axe-head-shaped in dorsal view due to the notch for the enlarged fourth dentary tooth in the lateral margin (figure 1*b*), in contrast with the paddle shape of *Nannosuchus* [8,26,27,33]. The naris is placed

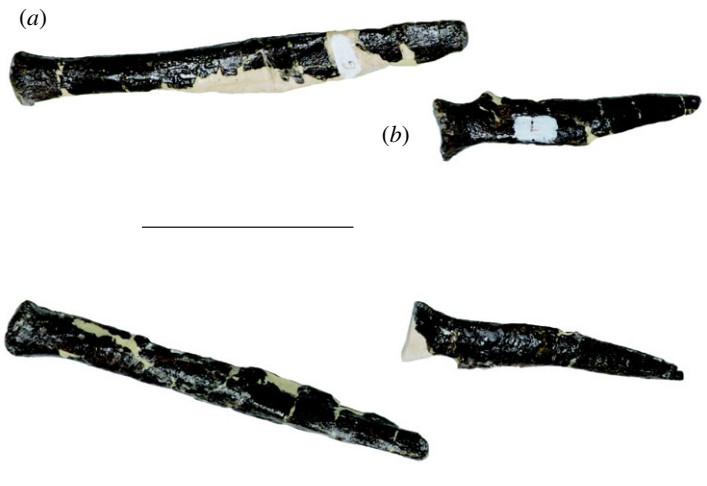

**Figure 8.** Pairs of ribs of the atlas (*a*) and axis (*b*) of *Amphicotylus milesi* (GMNH-PV0000229). Scale bar = 5 cm.

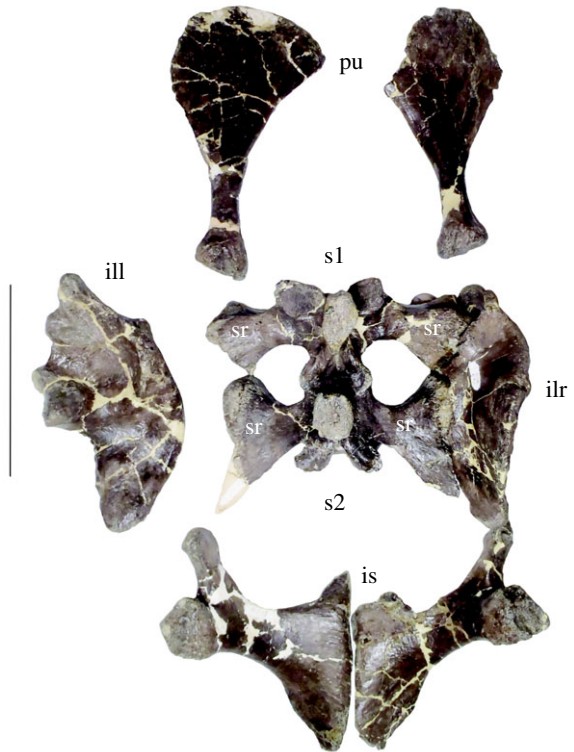

**Figure 9.** Sacrum and pelvic girdle of *Amphicotylus milesi* (GMNH-PV0000229) in dorsal view. Abbreviations: s, sacral vertebra; sr, sacral rib; pu, pubis; il(r and l), ilium (right and left); is, ischium. Scale bar = 10 cm.

anteriorly and entirely encircled by the premaxillae as in some goniopholidids such as *Eutretauranosuchus*, *Sunosuchus*, *Amphicotylus lucasii*, *G. simus* and *G. kiplingi* [8]. The anteriorly positioned external narial opening is circular in outline and anteroposteriorly short (approx. 50% the length of axe-head region of rostrum), when compared to *G. kiplingi* [8]. Lateral to the contact between the anterior rami of the premaxillae, there is a shallow fossa as in *Anteophthalmosuchus hooleyi* and *Amphicotylus* [8], but not as deep as that of *G. kiplingi* and *G. simus* [8]. The anteronarial foramen is present on the dorsal surface between the first premaxillary tooth and the naris and is similar to the anteronarial notch of *G. kiplingi* [8]. A perinarial crest surrounding both lateral and posterior margin of the naris is barely developed, and the postnarial fossa is absent, differing from *G. kiplingi* [8]. The posterior margin of the naris projects anteriorly, resulting in a figure-eight narial morphology as seen

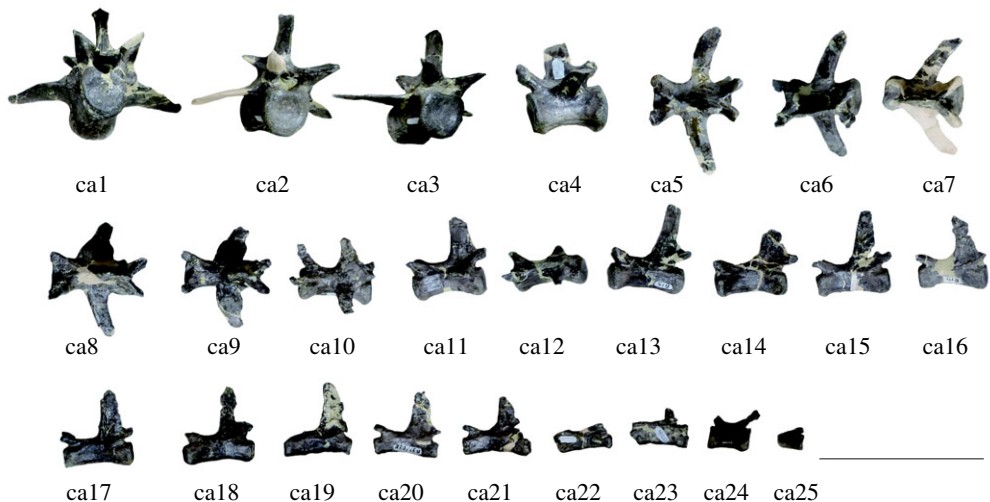

**Figure 10.** Caudal vertebra of *Amphicotylus milesi* (GMNH-PV0000229): ca1–3 in anterior, ca4 and 11–25 in lateral, ca5–10 in dorsal views. Scale bar = 10 cm.

in *Amphicotylus lucasii* and *Amphicotylus stovalli* [26,27,34]. On the dorsal surface, an anteronarial foramen is present anterior to the naris, similar to *G. kiplingi* [8]. The premaxilla bears five alveoli, although the specimen lacks the first right and the last left erupted teeth (figure 1*c*). The third and fourth premaxillary teeth are the largest of the premaxillary and maxillary teeth. Two ventral pits are present on the ventral surface of the premaxilla.

The maxillary depression is present on the laterodorsal surface of the posterior process (figure 1*d*), as in the other goniopholidids and pholidosaurids [35]. The number of subfossae in the depression is five (figure 4) as in *G. simus* [36,37], while it is four in *Amphicotylus lucasii* and three in *G. kiplingi* [8] and *Anteophthalmosuchus hooleyi* [35]. A neurovascular opening is present at the bottom of each chamber in *G. kiplingi* [8]; however, it is only present in the second chamber of the left depression of *Amphicotylus milesi* (GMNH-PV0000229). The posterior margin of the nasals is divided by the frontal as in *G. kiplingi* [8] and *Anteophthalmosuchus hooleyi* [35].

The lacrimal (figure 1*b*) is as long as the prefrontal and the ratio of its anteroposterior length to its mediolateral width is approximately 3.0, which is less than that of *G. kiplingi* (approximately 5.0) [8]. A lacrimal groove is present in the dorsal surface lateral to the periorbital crest. The prefrontal is longer than wide and similar in size and shape to the lacrimal in dorsal view. A ventral ramus is not preserved and the condition of prefrontal pillar is unknown.

In the frontal, the width of the anterior process (39 mm) is larger than that of the posterior process (24 mm) like other goniopholidids such as *Amphicotylus stovalli*, *G. simus*, *Anteophthalmosuchus* and *Nannosuchus* [8,34,35,37,38]. The frontal-postorbital suture is almost straight and simple unlike *G. simus* and *G. kiplingi* [8,37] which is tenon-shaped. The frontal-parietal suture is visible at the intertemporal bar in lateral view, but not in dorsal view.

The well-sculpted, single parietal forms the posteromedial margin of supratemporal fenestra (figure 1*b*). The dorsal surface is flat and three times as wide posteriorly as in the intertemporal bar, which is larger than that of *Anteophthalmosuchus hooleyi* [35].

The ornamented postorbital is triradiate and composed of medial, posterior and ventral rami. A very short and robust process projects anterolaterally from the anterolateral corner of the postorbital, as in *G. simus* and *G. kiplingi* [8]. Its contacts with the frontal and palpebral are straight and simple, but its contact with the squamosal is obscured by the fusion of the elements.

The squamosal contacts the parietal medially and the postorbital anteriorly (figure 1*b*). The ornamented dorsal surface of the squamosal extends posterolaterally over the anterior portion of the quadrate as a horizontal lobe that gives the skull table its sinusoidal profile in dorsal view.

The jugal is triradiate with anterior, posterior and ascending rami. Both the anterior and posterior rami are flattened and ornamented, whereas the ascending ramus is cylindrical in cross-section and unornamented. The anterior ramus contacts with the lacrimal medially and the maxilla dorsal to the maxillary depression. A dorsal crest extends anteroposteriorly on the medial margin of the anterior ramus (figure 1*b*). The posterodorsal corner of the anterior ramus is dorsally concave and participates

**Figure 11.** Appendicular skeletons of *Amphicotylus milesi* (GMNH-PV0000229). (*a*) Left shoulder girdle in lateral view. Left forelimb in anterior (*b*) and posterior views (*c*). (*d*) Right manus in dorsal view. (*e*) Interclavicle in ventral view. (*f*) Left pelvis in lateral view. (*g*) Left femur in medial view and fibula and tibia in anterior view. (*h*) Right tibia, fibula and pes in anterior view. Abbreviations: sc, scapula; co, coracoid; hu, humerus; ra, radius; ul, ulna; ule, ulnare; rae, radiale; mc, metacarpal; mu, manual ungual; ic, interclavicle; pu, pubis; il, ilium; is, ischium; fe, femur; fi, fibula; ti, tibia; as, astragalus; ca, calcaneum; mt, metatarsal; pu, pedal ungual. Scale bar is 10 cm.

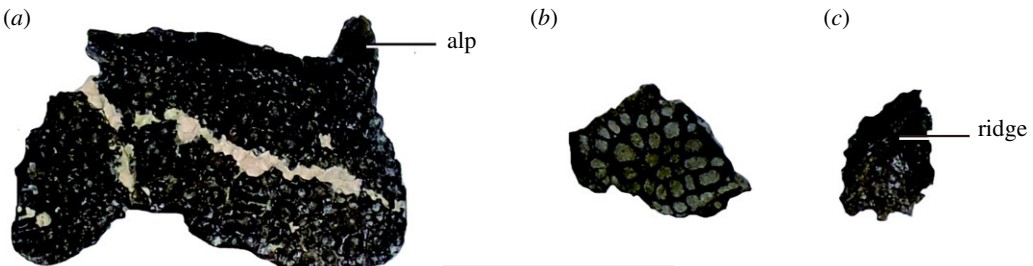

**Figure 12.** Osteoderms (GMNH-PV0000229) of the dorsal shields (*a*), ventral shields (*b*) and accessories (*c*). Abbreviations: dso, dorsal shield osteoderms; vso, ventral shield osteoderms; ao, accessory osteoderms. Scale bar is 5 cm.

in the anterior margin of the orbit. This preorbital fossa is bordered by the dorsal crest of the jugal and the upper periorbital crest that is composed of the lacrimal and prefrontal. The posterior ramus contributes medially to the ventralmost margin of the infratemporal fenestra and extends beyond the posterior end of the infratemporal fenestra laterally to contact the quadratojugal. The dorsal margin of

the jugal extends anterior to the orbit and flares laterally at the lacrimal-jugal suture, forming the lower periorbital crest at the anterolateral margin of orbit.

The ornamented quadratojugal body is firmly attached to the jugal and quadrate, and the sutures with these elements are difficult to recognize due to their fusion (figure 1b,d). The unornamented ascending spine is narrow and bounds most of the caudal margin of the infratemporal fenestra. The quadratojugal spine extending from the anterior edge of the ascending process is present in G. simus [37], but not preserved in Amphicotylus milesi (GMNH-PV0000229). The posterior end of the quadratojugal contacts the lateral surface of the quadrate, but does not contribute to its articular facet for the mandible.

The robust, unsculptured quadrate is oriented posterodistally (figure 1b). Its smooth anterodorsal surface forms most of ventral surface of the otic recess. A preotic siphonal foramen is present anterior to the otic recess. There are two well-developed crests on the ventral surface of the quadrate: crest A extends the length of the quadrate-quadratojugal suture; crest B is positioned medially to crest A and extends posteromedially parallel to the quadrate-pterygoid suture (sensu Iordansky [39]). The distal condyles of the quadrate are positioned posterior to the occipital condyle and subdivided into lateral and medial portions by a median groove. The medial portion is dorsoventrally higher than the lateral portion, in contrast with G. baryglyphaeus [40].

The exoccipitals are fused to each other dorsal to the foramen magnum (figure 1g). The exoccipitals form the anterior and the lateral walls of the foramen magnum, whose ventral wall is bounded by basioccipital. The exoccipital forms the caudally flexed paroccipital process. A carotid foramen opens ventrolaterally on each side of the occipital surface in the ventromedial region of the exoccipitals. Due to damage, the foramen vagus and the suture between the exoccipital and supraoccipital are unrecognizable.

The supraoccipital is excluded from the skull table by the parietal bordering it dorsally, and its lateral and ventral margins contact the exoccipitals (figure 1g). The outline of the supraoccipital is unclear due to the multiple cracks running through the occipital surface. The supraoccipital lacks a median crest on the caudal surface, differing from G. baryglyphaeus [40].

The basioccipital forms the occipital condyle and contacts the exoccipitals dorsolaterally (figure 1g). The occipital condyle has a sub-circular articular facet. A shallow median sulcus extends into the foramen magnum on the dorsal surface of the occipital condyle. Ventral to the occipital condyle, the basioccipital plate is concave and three small foramina, interpreted as the eustachian tubes, are transversely positioned.

The basisphenoid is positioned ventral to the basioccipital, contacts the basioccipital, pterygoid and quadrate and is visible in lateral view. The ventral surface of the basisphenoid has reduced exposure on the ventral surface of the braincase.

The long, smooth ectopterygoid is attached anteroposteriorly to the lateral margin of the pterygoid wing. Due to the crushing, its contacts with the postorbital, jugal and maxilla are obscure.

The laterally broadened mandibular symphysis is composed of the dentary and splenial (figure 1e). The splenial covers the posterior one-fifth of the mandibular symphysis as in other goniopholidids (e.g. Eutretauranosuchus [31] and G. baryglyphaeus [40]). Based on the number of alveoli, the dentary originally had 22 teeth on each side, but only 10 (left) and 20 (right) teeth are preserved. The alveolar margin is not bordered by the splenial. The medial surface of the splenial is smooth and possesses a Meckelian groove.

The angular forms a trough that would have enclosed the Meckelian cartilage. The angular possesses a well-developed ridge ventral to the suture with the articular.

The surangular contributes to the posterodorsal portion of the mandible. Anterior to the glenoid fossa, the dorsal edge of the surangular is straight. The external surface is well-ornamented with circular pits. The anterior portion of the surangular continues dorsally along with the posterolateral margin of the last three to four alveoli.

The external mandibular fenestra is small, bordered by the angular and surangular, and is longer than wide with an acute posterior corner (figure 1e).

The articular forms the posterior margin of the medial mandibular fossa (figure 1e). The mandibular glenoid is more than twice as wide as mandibular ramus and is divided into lateral and medial portions, corresponding to the quadrate articular facet. The dorsal surface of the retroarticular process is concave and smooth.

The dentition is homodont both in the upper and lower jaws (figure 1d,e). The tooth crown is sub-circular or elliptical in cross-section. The lingual and labial surfaces are asymmetric. The labial surface is strongly arched, while the arching of the lingual surface is less pronounced. Low apicobasal ridges are present on both the lingual and labial crown surfaces. A distinct keel extends along with both the mesial and distal sides of each tooth crown.

A pair of ceratobranchials was recovered (figure 1*f* and 5). The left ceratobranchial is positioned ventral to the pterygoid (figure 5) and is relatively complete, although it lacks the dorsal margin of the distal end, while the right ceratobranchial is missing its distal portion. The left ceratobranchial has a total preserved length of 99 mm along with the dorsal margin and has anapproximately 35% skull width, which is less than *Euparkeria* [41], *Aplestosuchus* [42] and the paralligatorid *Batrachomimus* [43] which are over 50% skull width, but is as long as in modern crocodylians (figure 6*a*). The Wilcoxon rank sum test shows a significant difference between neosuchians and non-neosuchians in the relative size of the ceratobranchial to skull width ($p = 0.0009$) with the ceratobranchial of neosuchians (goniopholidids, paralligatorids, and eusuchians) being shorter than that of non-neosuchians (figure 6*a*). The left ceratobranchial is dorsally deflected along with their length. The ratio of the length of the left ceratobranchial along with the dorsal margin to the shortest distance between the proximal and distal ends is 1.31, which is as high as in paralligatorid *Batrachomimus* [43] and eusuchians (figure 6). The distal portion is narrower than the proximal portion and is slightly bowed medially. Near the midpoint of the curvature, there is a prominent tubercle in the ventromedial surface, as seen in *Goniopholis* sp. [44]. The posterior one-fifth of the proximal portion increases in height towards its terminus.

Twenty-four presacral vertebrae comprise nine cervical, ten thoracic and five lumber vertebrae (figures 7–10), and the centra are amphicoelous without hypapophyses. Two pairs of the sacral ribs are not fused with the ilia. The 25 preserved caudal centra are weakly amphicoelous or amphiplatyan.

Cervicals: The proatlas is missing, but the unfused, paired atlantal neural arches are preserved (figure 7), as well as atlantal ribs (figure 8). The neural arch is a plate-like process, and its dorsal margin extends medially. The medial surface of the ventral portion of the atlantal neural arch is concave. The atlantal rib is approximately twice as long as the axial rib (figure 8). The atlantal rib is the longest of the cervical ribs, with a rod-like proximal and a planar distal section.

The axis is completely preserved, including the axial ribs (figures 7 and 8). The centrum sutures with both the neural arch and the odontoid are incompletely fused. The parapophysis is developed on the lateral surface of the odontoid, making it wider than the body of the axial centrum. Posterior to the parapophysis, the lateral surface of the axis has a small fossa. The centrum is similar in length to the post-axial cervical vertebrae. It is medially constricted in dorsal and ventral views. The axial neural arch is slightly taller than the centrum. The neural spine is thin and anteroposteriorly broad, extending the entire length of the neural arch. The dorsal margin of the axial neural spine is horizontal and extends posteriorly over the postzygapophyses. The postzygapophyses form an obtuse angle, separated by a fossa. The prezygapophyses are small, but appear to be complete. The neural canal is higher than wide. The axial rib is planar and has an ascending crest on the dorsal margin of the proximal portion (figure 8*b*). Distal to the dorsal crest, the dorsal margin is straight, but the ventral margin is dorsally oriented in the distal portion, resulting in the rib tapering to its terminus.

The third to seventh cervical vertebrae possess anteroposteriorly long diapophyses and parapophyses, being positioned dorsoventrally higher on the eighth and ninth cervical vertebra (figure 7). In addition, the diapophyses project ventrolaterally on the third to seventh cervical vertebrae, and horizontally on the eighth and the ninth cervical vertebrae as they do in the dorsal vertebrae. The diapophyses shift further dorsally after the seventh cervical vertebra. The articular surface of the diapophysis for the cervical rib is convex. The neural spine becomes taller in the posterior cervical vertebrae, reaching its maximum height in the last cervical vertebrae. The anteroposterior length of the cervical neural spines remains constant except for that of the third vertebra that expands distally. Posteriorly, the length of the post-axial cervical centra becomes gradually larger and the angle of the prezygapophyseal and postzygapophyseal articular facets becomes steeper.

Dorsals: The 15 dorsal vertebrae can be separated into two segments, thoracic (first to tenth dorsals) and lumbar (eleventh–fifteenth dorsals) (figure 7). The neural spines of the thoracic vertebrae are almost vertical, while those of the lumbar vertebrae are inclined anteriorly. The articular surfaces of the centrum are oval with a transversely oriented long axis. The first and second dorsal vertebrae resemble the last two (eighth and ninth) cervical vertebrae in having their parapophyses located on the centrum, with both the diapophyses and parapophyses being taller than wide. The first dorsal vertebra does not preserve its diapophysis. The second dorsal vertebra differs from the last two cervical vertebrae in having a distinctively wider transverse process. The transverse processes of the third thoracic through second lumbar vertebrae are wider than the first and second thoracic and third through fifth lumbar vertebrae. The transverse processes are rounded in cross-section in the first and the second dorsal

vertebrae, and are dorsoventrally flatter on all other dorsal vertebrae. The thoracic centra are higher than wide, while the lumbar centra are wider than high. The lateral surfaces of the dorsal centra are smooth, except for the first and second dorsal vertebrae with their parapophyses.

Sacrals: The two preserved sacral vertebrae are unfused to each other, but do have fused sacral ribs (figure 9). They are almost complete except for the posterolateral process being missing from the left second sacral rib. The neurocentral suture is visible in ventral view on both sacral vertebrae. The neural spine is anteroposteriorly longer in the first sacral than the second sacral vertebra. The distal end of each neural spine is expanded transversely. The postzygapophysis of the first sacral vertebra and the prezygapophysis of the second sacral vertebra are narrower than the prezygapophysis of the first sacral vertebra and the postzygapophysis of the second sacral vertebra. The anterior facet of the first sacral vertebra is twice as wide as high and is the widest vertebrae in the skeletal column. The second sacral centrum has a concave posterior facet. The intervertebral articular facets between the first and the second sacral vertebrae are small and a sagittal groove extends through them. The anterior facet of the first sacral vertebra and the posterior facet of the second sacral vertebra are concave.

Caudals: Twenty-five caudal vertebrae are preserved although they do not appear to represent the complete caudal series (figure 10). The serial number of the caudals is uncertain. The twelve anterior caudals bear transverse processes, but they are missing on the thirteen posteriormost preserved vertebrae. The neural spines are narrow and high. A pathological tuberosity is displayed on the proximal portion of right transverse process, and on both processes, on the 8th and 9th largest caudal vertebrae, respectively (figure 10). The transverse processes project posterolaterally, and the length of their main axes gradually decreases posteriorly.

The preserved left scapula is 115 mm long and more than twice as long as the scapula-coracoid suture (48 mm) (figure 11a). The scapular blade projects medially and expands distally, being three times as wide as the shaft. The anterior margin is markedly concave and the posterior margin of the blade is straight. The developed acrominal ridge contributes to the anterodorsally oriented acrominal notch.

The coracoid is 103 mm long and slightly shorter than the scapula (figure 11a). The shaft is constricted and expanded medially. The anterior margin is less concave than the posterior margin. The coracoid foramen is located close to the scapula-coracoid suture and the glenoid. The lateral surface is shallowly concave ventral to the glenoid.

The unpaired, planar interclavicle lacks its posterior end (figure 11e). The preserved length of the interclavicle is 134 mm. The dorsal surface is slightly inflated and mediolaterally expanded at its midpoint.

The forelimb (humerus (21 cm) + ulna (17 cm) + metacarpal III (5 cm)) is as long as the hind limb (femur (21 cm) + tibia (16 cm) + metatarsal I (8 cm)). The humeri are each 21 cm in length, similar to the preserved femur (figure 11b,c). The humeral shaft is posteriorly curved with a circular cross-section. The humeral head is broad; however, the proximolateral margin is damaged and, thus, the point of origin of the deltopectoral crest cannot be determined. The mediodistal ventral surface of the humeral head is concave. The deltopectoral crest projects distally and bears a low, pointed tubercle. The distal humerus is divided into the lateral and medial condyles by the intercondylar groove. The articular surface of the lateral condyle articulates with the ulna and is larger than the medial condyle.

Both ulnae are preserved (figure 11b, c) and bowed in anteroposterior view. The ulna is 167 mm long and is approximately 80% of the length of the humerus (figure 11b,c). The proximal end is quadrangular. The olecranon process is slightly higher than the main proximal articular facet and is separated from it by a shallow groove. The distal end is posteriorly concave.

The radius is straight, 140 mm long, and approximately 85% the length of the ulna (figure 11d). The proximal end of the radius is sub-rectangular in the proximal view. The radius expands distally, and the distal end is longer than wide.

The radiale is 57 mm long, wider than high, and possesses a wide proximolateral process and three articular facets for the radius, the ulna and the ulnare (figure 11d). The radiale is subequal to the metacarpals in length. The proximal end of the radiale has a facet for the radius and is wider than its distal end, and is more expanded proximolaterally than proximomedially as in *G. simus* [36,37]. Posterolateral to the proximal end, the articular facet for the ulna is located on the proximal one-third of the radiale and faces posterolaterally, as in *Crocodylus*. Ventral to the facet for the radiale, the articular facet for the ulnare gradually grades into the posterolateral surface of the shaft. The distal end of the radiale is expanded, beveled laterally and rectangular in distal view.

The ulnare is 35 mm long, shorter than the radiale (approximately one-third of its length) and constricted at the shaft (figure 11d). The proximal (ulnar) articular facet is subtriangular in proximal view. The medial portion of the distal end faces medially for articulation with either the radiale or the distal carpal.

One right set of the metacarpals is preserved (figure 11*d*). Metacarpals I, II, III, IV and V are 33 mm, 47 mm, 48 mm, 46 mm and 28 mm long, respectively. Metacarpal I and V are shorter than the others and approximately one-third the length of the other metacarpals. The laterally concave metacarpal I is the most robust. The distal end of each metacarpal (except V) is slightly expanded and exhibits the extensor pit and collateral ligament fossae on the dorsal surface. The distal ends of metacarpals II to IV are separated into lateral and medial portions by an intercondylar groove.

Both ilia are completely preserved and are approximately 120 mm in length (figure 11*f*). The dorsal margin of the iliac blade is gently convex. The postacetabular process is as long as the acetabulum, projects horizontally and tapers to its terminus. Dorsal to acetabulum, a low supracetabular crest is present, which is the insertion site for the m. iliotibialis. There are two rugose articular surfaces medially where the sacral ribs contact. The ilium and ischium form the margins of the acetabulum (figure 11*f*).

The left pubis is completely preserved (figure 11*f*), but the right pubis lacks the medial side of the distal end. The pubis is excluded from the acetabulum. It is about 120 mm in length, slightly longer than the ischium and flattened along with its length except at the proximal end. The mid-shaft is constricted. The distal end is expanded in dorsal and ventral views. The medial margin is more concave than the lateral margin.

The ischium (figure 11*f*) is approximately 100 mm in length and shorter than the pubis. On the dorsal margin of the ischium there are two facets for articulation with the ilium. The posterior surface of the proximal portion of the ischial shaft is shallowly concave and more robust than seen in *Anteophthalmosuchus* [35]. The iliac peduncle is larger than the pubic peduncle.

Both femora are completely preserved (figure 11*g*). The left and right femora are 208 mm and 210 mm long, respectively, equal to that of the humerus, and sigmoidal in lateral and medial views. The femoral head is round. The fourth trochanter is well developed. The anterior surface adjacent to the distal margin is shallowly concave. The distal surface is divided into lateral and medial condyles by an intercondylar groove. Both distal condyles project posteriorly.

The left tibia is 155 mm long, equalling the length of the fibula and is nearly three-quarters of the femoral length (figure 11*g*). The proximal end of the tibia has lateral and medial condyles separated by a groove. Both of the condyles are equally developed. The proximal articular facet is concave. The posterolateral side of the proximal end is concave and has an articular facet for the fibula.

The fibula is transversely flat and 156 mm long (figure 11*g*). There is a tuberosity on the lateral surface 40 mm from the proximal margin. The medial surface of the distal portion of the fibula is flat and triangular in profile for contact with the tibia. The distal margin has a rounded articular surface for articulation with the tarsals.

The astragalus has a medial projection for the articulation with the calcaneum (figure 11*h*). Medially, the astragalus possesses a large, rounded head.

The calcaneum is 41 mm long and laterally concave to accommodate the astragalus (figure 11*h*). The anterior body has a flat ventral surface and convex dorsal surface, and articulates with the pedal elements. The posterior tuber has a deep groove. The anterior condyle and the posterior tuber are well separated by a notch. The astragalar facet is located medially to the anterior condyle.

The metatarsals are completely preserved (figure 11*h*), but only one pedal ungual is preserved. The metatarsal I, II, III and IV, is 84 mm, 80 mm, 70 mm and 71 mm long, respectively.

Forty-nine isolated osteoderms (figure 12) display three typical goniopholidid dorsal profile morphotypes that can be repositioned as follows: 22 rectangular osteoderms from the median paravertebral dorsal shield, 19 trapezoidal and hexagonal osteoderms from the ventral shield, and 8 circular osteoderms as accessory osteoderms lateral to the median shields or dorsal to appendicular skeleton [32,35]. Some of the rectangular osteoderms possess an anterolateral process at their lateral edges as in goniopholidids [32,35,36,45,46] and pholidosaurids [47,48,49].

# 4. Discussion

## 4.1. Phylogenetic analysis

Our phylogenetic analysis produced 30 most parsimonious trees with a tree length of 2338 (figure 13). The strict consensus tree obtained from the analysis recovers *Amphicotylus milesi* as a member of Goniopholididae. Monophyly of *Amphicotylus milesi* and *Amphicotylus stovalli* is supported by four synapomorphies: anterior end of frontal is posterior to prefrontal (Ch142 (0)), palatal shelves of palatines are not fully in contact at the mid-line (Ch213 (0)), anterior border of internal naris is

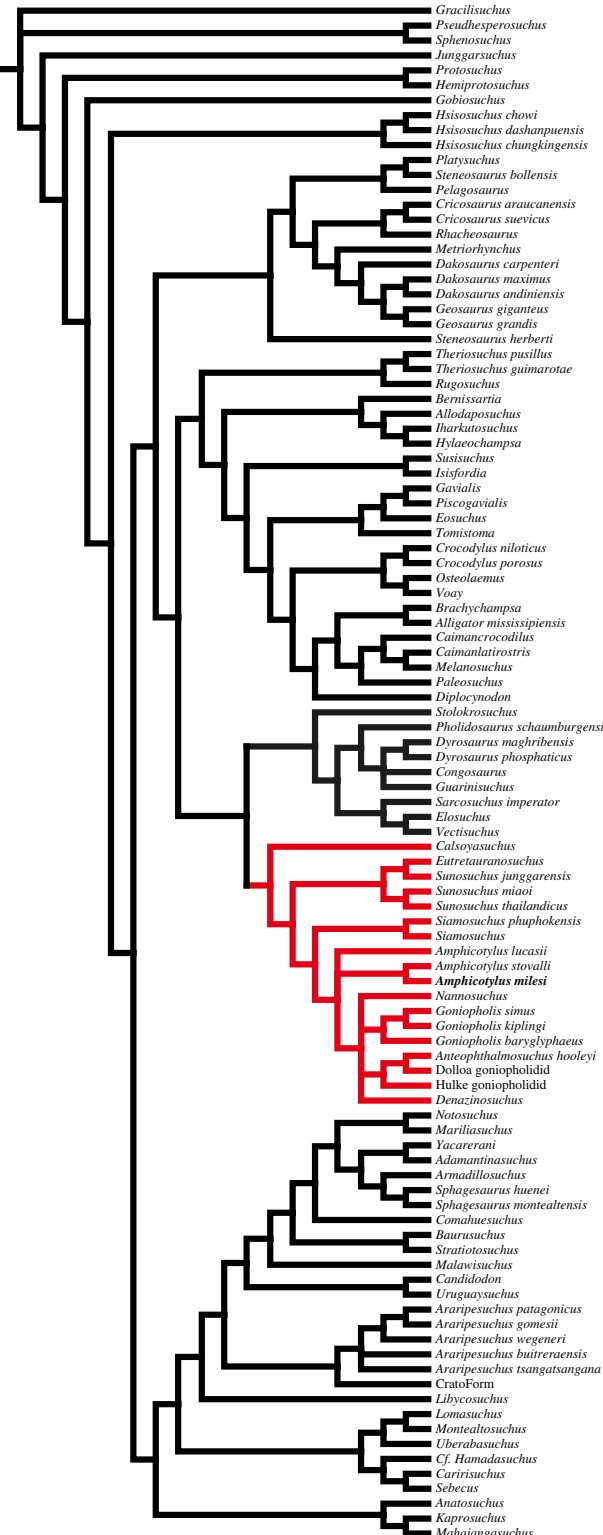

**Figure 13.** Consensus tree of the most parsimonious trees obtained. Red branches representing goniopholidids. *Amphicotylus milesi* forms a monophyly with *Amphicotylus stovalii*.

positioned anterior to suborbital fenestra (Ch235 (0)) and internal choana is lanceolated (Ch238 (1)). *Amphicotylus milesi* differs from *Amphicotylus stovalli* in having a broad rostrum, and lacking the crest C on ventral surface of quadrate, straight lateral border of nasals (Ch76 (1)), and rounded anterior margin of palatine (Ch225 (0)), where the latter two are autapomorphies of *Amphicotylus stovalli*. As in Andrade *et al.* [8] and Martin *et al.* [35], our analysis supports a monophyly of pholidosaurids and

goniopholidids, based on four synapomorphies: maxillary depression (Ch89 (1)), presence of both dorsal and ventral shields (Ch468 (2)), dorsal osteoderm with the anterolateral process (Ch477 (1)) and amphicoelous vertebra with centra slightly longer than tall (Ch429 (1)).

## 4.2. Evolution of internal choana in goniopholidids

The position of the internal choana is one of the important features for the early evolution of Eusuchia. Goniopholidids, which are close relatives of Eusuchia (figure 13), exhibit a variety of palatal element arrangements for the internal choana, indicating a complex evolution of choana in this group [12,13]. In two basal goniopholidids, *Amphicotylus milesi* and *Eutretauranosuchus*, the choana is anteroposteriorly elongate and bordered by the maxilla, palatine and pterygoid [13,31]. Another basal goniopholidid, *Sunosuchus*, has separated choanal openings anteriorly and posteriorly due to the contact of the palatines between them [32]. Two derived goniopholidids, *Anteophthalmosuchus* and *Hulkepholis*, have a short and posteriorly placed internal choana, bordered by the palatines and pterygoids, similar to the basal eusuchian *Isisfordia* [13,35,50].

The anterior extension of the internal choana is unique to basal goniopholidids and paralligatorids among neosuchians (figure 3a,b). The anterior margin of the internal choana of basal goniopholidids and paralligatorids lies above the upper quartile of neosuchians (figure 3a). The condition in *Eutretauranosuchus* and *Sunosuchus* is extreme, similar to that in the pseudosuchian *Ornithosuchus*. *Amphicotylus milesi*, as well as paralligatorids, is in the range of a quartile of non-neosuchians. The anterior margins of the internal choana in the derived goniopholidids (*Anteophthalmosuchus* and *Hulkepholis*) are placed within the quartile of neosuchians (figure 3a). Regardless of the various positions of the anterior margin of the internal choana in goniopholidids, the posterior margin of the internal choana is placed anteriorly in non-neosuchians and posteriorly in neosuchians, including goniopholidids (figure 3b,c), which is supported by the Wilcoxon rank sum test ($p < 0.0001$). The positions of the posterior margin of the internal choana of goniopholidids commonly fall within the quartile of neosuchians (figure 3b,c).

The quantitative measurements reveal that in the basal goniopholidids, *Amphicotylus*, *Eutretauranosuchus* and *Sunosuchus*, the internal choana is uniquely open both anteriorly and posteriorly in the palate. However, the anterior portion of their internal choana is inferred to have been covered by soft tissue as in the soft pterygoid secondary palate of the *Alligator* embryo in which an epithelium covers the unfused palatines between 24 and 44 days after egg laying [51,52]. Therefore, the covered palate of goniopholidids could have been a shared feature with modern crocodylians, as are the other regions of the oral cavity that are covered by soft tissue (i.e. suborbital fenestra and incisive foramen) [39].

## 4.3. Ceratobranchial size and curvature

The ratio of ceratobranchial length to mandibular width of *Amphicotylus* is 0.35 is as short as it is in all neosuchians except *Batrachomimus* (figure 6a). A wide variation in ceratobranchial size may be a plesiomorphic condition in crocodyliforms (figures 6 and 16), while neosuchians have a short ceratobranchial exhibiting a limited disparity in size except in the paralligatorid *Batrachomimus* (figure 6a). A short ceratobranchial in neosuchians indicates (i) the posterior shift of the hyoid apparatus in which the basihyal encases a larynx in the pharynx [3], (ii) the relative reduction of the pharyngeal region and (iii) an increase of the oral region for holding and crushing of food items. Regardless of the instability for the phylogenetic position of thalattosuchians (the sister group of Neosuchia in this study) in recent analyses [39,53], our results (figure 6a) confirm that oral enlargement had already occurred in Neosuchia or more the basal clade Thalattosuchia + Neosuchia.

*Amphicotylus* shows that the dorsal deflection of the ceratobranchial posterior ramus, as seen in paralligatorids and eusuchians, occurred after oral enlargement in crocodyliforms. The Wilcoxon rank sum test shows a significant difference between neosuchians and non-neosuchians in this curvature (figure 6b, $p < 0.0001$), indicating it originated at the base of Neosuchia (figure 16).

## 4.4. Hyoid musculature of modern crocodylians and Amphicotylus milesi

Previous studies [6,7,54] have illustrated five hyolingual muscles (m. hyoglossus, m. branchiomandibularis rostralis and caudalis, m. episternobranchialis, and m. omohyoideus) attached to the ceratobranchials of crocodylians, but dissections of two modern crocodylians

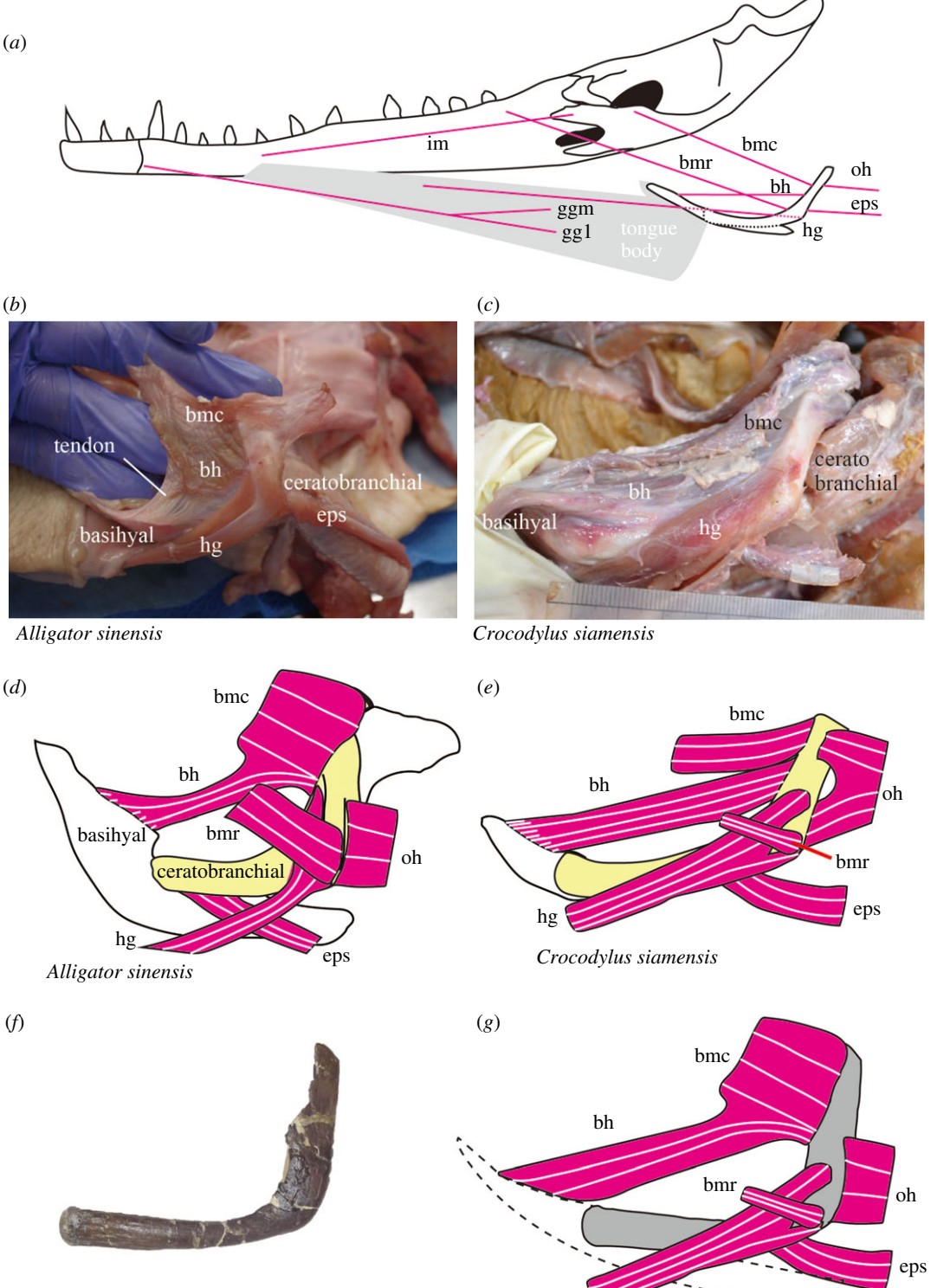

**Figure 14.** Schematic illustration of mandible, hyoid apparatus and hyolingual musculature in crocodylians. Muscular attachment sites in *Crocodylus siamensis* (*a*). Pictures of hyolingual musculature and its schematic illustration in *Alligator sinensis* (*b,d*) and *Crocodylus siamensis* (*c,e*). Ceratobranchial of *Amphicotylus milesi* (GMNH-PV0000229) (*f*) and its reconstruction (*g*). Abbreviations: im, intermandibularis; bh, branchiohyoideus; bmr, branchiomandibularis rostralis; bmc, branchiomandibularis caudalis; oh, omohyoideus; eps, episternobranchialis; hg, hyoglossus; ggm, genioglossus medialis; ggl, genioglossus lateralis.

(*Crocodylus siamensis* and *Alligator sinensis*) in this study newly reveal that the sixth hyolingual muscle, m. branchiohyoideus, is present (figures 14 and 15) as in other reptiles such as turtles and lizards [55,56]. These hyolingual muscles are involved in two major movements: protraction-suspension and

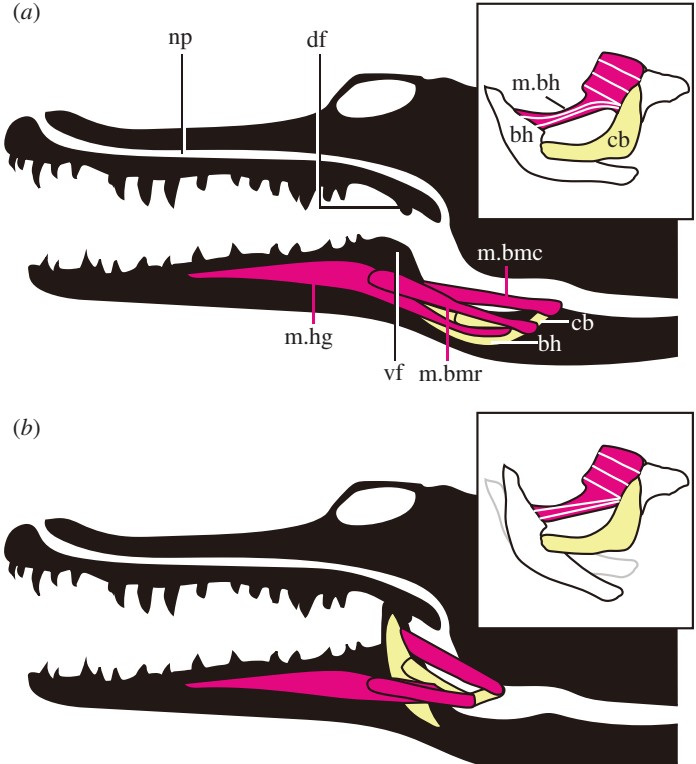

**Figure 15.** Schematic illustration of open-closure mechanism of gular valve in *Alligator sinensis* with hyoids and four hyolingual muscles in valve open condition (*a*) and close (*b*). Function of m. branchiohyoideus is shown in the box. Abbreviations: cb, ceratobranchial; bh, basihyal; df, dorsal fold of gular valve; np, nasopharyngeal passage; m. bh, musculus branchiohyoideus; m. bmc, musculus branchiomandibularis caudalis; m. bmr, musculus branchiomandibularis rostralis; m. hg, musculus hyoglossus; vf, ventral fold of gular valve.

retraction-depression of the hyoid (basihyal-ceratobranchial-complex) and dorsiflexion of the basihyal with respect to the ceratobranchials. The hyoid is protracted and suspended by m. hyoglossus, m. branchiomandibularis rostralis and caudalis and retracted and depressed by m. omohyoideus and m. episternobranchialis [6,7,54]. M. branchiohyoideus, originating from the dorsal margin of posterior portion of the curved ceratobranchial and inserting onto the lateral margin of the anterior portion of the basihyal, pulls the basihyal posterodorsally (figure 15).

The general morphology, especially the curved nature of the ceratobranchial and the muscle attachments (e.g. ventral tubercle for m. hyoglossus), of the ceratobranchials of *Amphicotylus milesi* is similar to those of modern crocodylians; therefore, the arrangements of the hyolingual muscles of *Amphicotylus milesi* are inferred to be comparable to modern crocodylians. Osteological correlates for m. branchiohyoideus are not preserved in *Amphicotylus milesi*; however, the muscle was likely present as in modern crocodylians (this study), turtles [55] and lizards [56].

The dissection of the hyoid apparatus demonstrates that the m. branchiohyoideus can pull up the anterior portion of basihyal to raise the ventral fold of the gular valve (figure 15) thereby dividing the oral and pharyngeal cavities [3]. The dorsal deflection of the ceratobranchial in *Amphicotylus milesi* increases the moment arm of this muscle. This deflection is present in both extinct and extant neosuchians (figure 6*b*), which were likely capable of rotating the basihyal as in modern crocodylians (figure 15). On the other hand, non-neosuchian crocodyliforms have nearly straight ceratobranchials (figure 6*b*) which would have prevented any significant rotation of the basihyal to raise the ventral fold of gular valve.

Our recognition of the morphology of fossil hyoid apparatus and the plausible reconstruction of the muscular anatomy of *Amphicotylus milesi* in this study allows us to conclude for the first time that this taxon and other extinct neosuchians could have raised the ventral fold of the gular valve in a similar manner to modern crocodylians allowing them to breath while opening the mouth underwater when their external nares were above the water surface. The recognition of this key innovation newly sheds some lights on semi-aquatic adaptation in crocodyliform evolution.

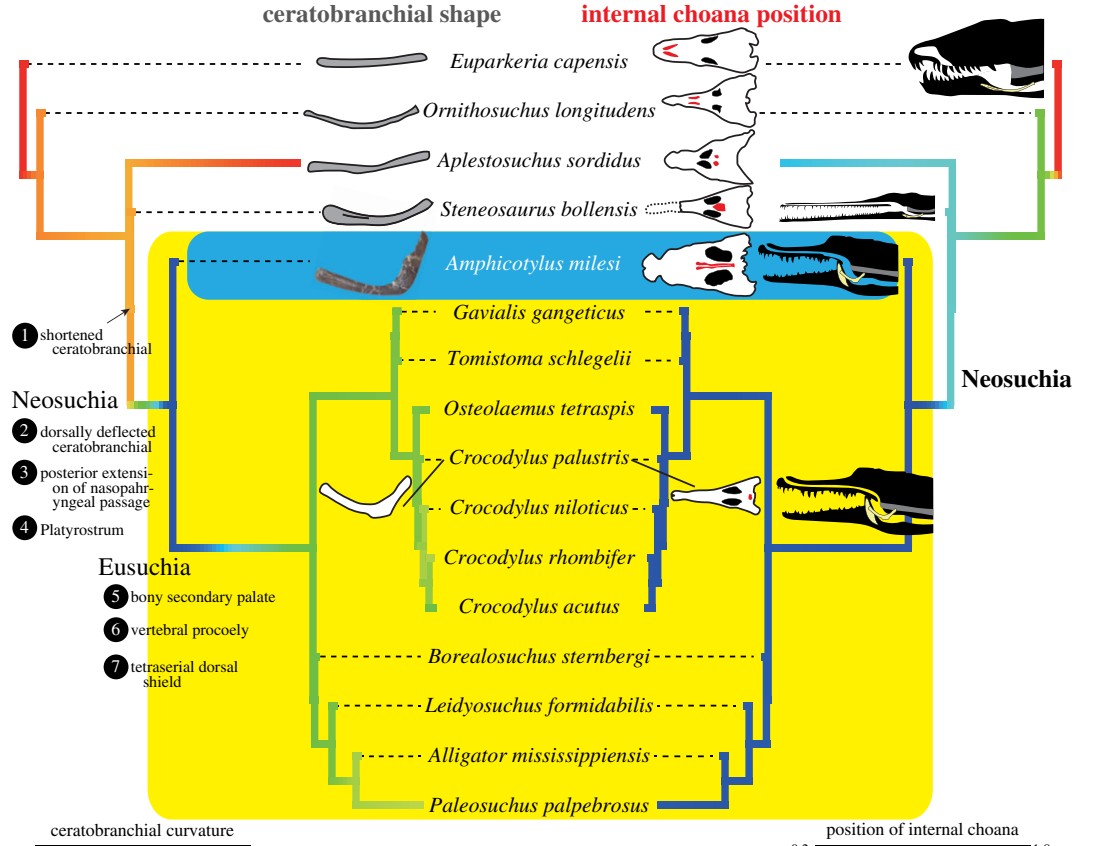

**Figure 16.** Phylogenetic relationships of Goniopholididae and evolutionary history of ceratobranchial (left) and internal choana (right) in Crocodyliformes. Lateral side of ceratobranchial and ventral side of skull are illustrated. Schematic drawings (silhouette) on the right side show hyoids (yellow), respiratory duct (transparent) and esophagus (grey), depicting evolutionary process of acquisition of neosuchian respiratory system that raise ventral fold of gular valve.

## 4.5. Aquatic adaptation in the palate and hyoid

Our analyses of the ceratobranchial curvature and the internal choana demonstrate that neosuchians, including the goniopholidid *Amphicotylus milesi*, had acquired a modern crocodylian-type respiration involving separation of the narial and oral passages through the action of a gular valve (figure 16; electronic supplementary material, S1–3), [2,23,50,52] at the beginning of their evolutionary history. Late Jurassic goniopholidids such as *Amphicotylus* had a crocodylian-like cranium with a platyrostrum and a pterygoid secondary palate, as well as a functional gular valve. Middle Cretaceous to present derived eusuchians have retained a complete bony pterygoid secondary palate and evolved vertebral procoely, a great range of mediolateral vertebral motion and flexible tetraserial dorsal shields [50,57,58,59]. Therefore, within neosuchian evolution, the respiratory modifications of the cranium and development of a functional gular value that allowed for a semi-aquatic lifestyle occurred during the Early to Middle Jurassic, well in advance of the locomotive adaptations in postcranium seen in younger, more derived eusuchians during the Cretaceous and later (figure 16; electronic supplementary material, S1–3).

Ethics. Material described in this paper was legitimately collected and adhere to the Society of Vertebrate Paleontology's guidelines.

Data accessibility. All data and code used in the study is provided in the electronic supplementary material. This published work and nomenclatural acts have been registered in ZooBank, the proposed online registration system for the International Code of Zoological Nomenclature. The ZooBank life science identifiers (LSID) can be resolved and the associated information viewed by appending the LSID to the prefix http://zoobank.org/. The LSID for this publication is urn:lsid:zoobank.org:pub:3191C7AA-9786-489E-AFB9-EF1D6F4DD002.

    The data are provided in the electronic supplementary material [60].

Authors' contributions. J.Y. and A.H. designed the study, collected data and performed the comparative and analytical work. J.Y., A.H., Y.K., M.J.R., Y.T. and Y.H. provided data and contributed to writing and discussion.

Competing interests. We declare we have no competing interests.

Funding. This work was supported by the Sasakawa Scientific Research Grant of the Japan Science Society and JSPS Overseas Challenge Program for Young Researchers and KAKENHI grant no. 21K14026 to J.Y.

Acknowledgements. We thank Western Paleontological Laboratories, Inc. for excavation and preparation. We express gratitude to staffs of Gunma Museum of Natural History and Takashi Oda (Seian University of Arts and Design) for technical assistance. We also thank Carl Mehling (American Museum of Natural History) and Max Langer (Universidade de São Paulo) for providing specimen access.

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
