## [Peer Review File · Royal Society Open Science]

Review History

RSOS-210320.R0 (Original submission)

Review form: Reviewer 1

Is the manuscript scientifically sound in its present form?

Yes

Are the interpretations and conclusions justified by the results?

Yes

Is the language acceptable?

Yes

Do you have any ethical concerns with this paper?

Yes

Have you any concerns about statistical analyses in this paper?

No

Recommendation?

Accept with minor revision (please list in comments)

Comments to the Author(s)

Great work on a wonderful specimen.

Review form: Reviewer 2 (Márton Rabi)

Is the manuscript scientifically sound in its present form?

Yes

Are the interpretations and conclusions justified by the results?

Yes

Is the language acceptable?

Yes

Do you have any ethical concerns with this paper?

No

Have you any concerns about statistical analyses in this paper?

No

Recommendation?

Major revision is needed (please make suggestions in comments)

Comments to the Author(s)

Dear authors,

This is an impressive study and an essential contribution to our understanding of aquatic adaptations in neosuchian crocodyliforms. I placed my comments into the attached pdf (see Appendix A), they are minor but given the number of them I recommended "major revision". Overall, I suggest including a larger figure of the skull of the specimen. The material certainly deserves larger resolution photos and page restrictions no longer apply. Similarly, the authors could illustrate some of their observations derived from the dissecting besides the schematic representation.

I disagree with some details of the conclusions: given the instability of thalattosuchia in phylogenies and the poor sampling of ceratobranchials in the tree makes it difficult to trace the origin of the gular valve closing. Moreover, while the hypothesis of the authors regarding the function of the curved ceratobranchial makes perfect sense, it is a hypothesis not a fact (only a single event in the phylogeny) and the evolutionary implications should be proposed with less certainty.

I am looking forward to see this excellent and essential work to be published!

Review form: Reviewer 3

Is the manuscript scientifically sound in its present form?

No

Are the interpretations and conclusions justified by the results?

No

Is the language acceptable?

No

Do you have any ethical concerns with this paper?

No

Have you any concerns about statistical analyses in this paper?

Yes

Recommendation?

Major revision is needed (please make suggestions in comments)

Comments to the Author(s)

First of all, I'd like to apologise for my review being late. Secondly, I do like the 'thrust' of this manuscript, I just think it requires major modifications to be acceptable for RSOS.

Odd statements

There are some odd comments here and there, such as this from the introduction:

"In the evolutionary history of crocodyliforms, the presence of the secondary palate is a key character for eusuchian crocodyliforms, and it appeared in the successive stem groups of Eusuchia such as paralligatorids and derived goniopholidids "

All mesoeucrocodylians have a maxillopalatine secondary palate, and there is an independent maxillopalatine secondary palate in Shartegosuchoidea (even an independent acquisition of the 'pterygoid secondary palate'). I'm not entirely sure what the authors are meaning here.

Another one is:

"Goniopholidids, which is a transitional group to Eusuchia,"
Goniopholidids aren't. It's an odd thing to say.

And there are more of these. I think the authors need to give a serious re-think to the tone of the manuscript. It reads very teleological, with neosuchian groups being 'pit-stops' on the road to Eusuchia. They weren't, many, such as goniopholidids, were evolutionary successful lineages that persisted for tens of millions of years.

Phylogeny

As the authors will no doubt be aware crocodylomorph phylogeny is chaotic in places. The position of major clades are in flux. So the comment:

"Goniopholididae is a group of basal neosuchian crocodyliforms found in the in the northern hemisphere during the Jurassic and Early Cretaceous, and a sister clade to Paralligatoridae and Eusuchia"

Needs to be better cited. I would simply state that Goniopholididae is a successful lineage of neosuchians outside of the Paralligatoridae + Eusuchia clade. That way, what you are saying doesn't require discussing competing phylogenetic hypotheses.

Also, there is a completely different topology used for the R analyses in the SI... Where does it come from? Why are you not using the topology you analysed with the modified Andrade dataset?

New species

As RSOS is an online-only journal can the authors make sure everything they do is done in accordance with the ICZN Code? That means ensuring at proof stage the ZooBank LSIDs for both the paper and the new species are present.

Anatomy and SI

So, the anteriorly elongated internal nares. I can't see it in the figures. There are no close-ups, only interpretative line drawings. And from those, I would hesitate to say the morphology isn't purely post-mortem damage.

To be honest, the description and figuring are what frustrate me most. As the authors say, this is the most complete known skeleton of a goniopholidid, and the description is largely in the SI and so are the figures. The complete description needs to be 'main text', there needs to be much more, and better figures.

I'm really sorry, but it's really not acceptable to 'hide away' the vast bulk of the description, and to give such limited figuring. There are some really good figures in the SI, but they are put into multi-element plates that reduce their size and quality (when you zoom in to double check the anatomy). The SI figures should be split into smaller units, and added into the main document.

Decision letter (RSOS-210320.R0)

Dear Dr Yoshida

The Editors assigned to your paper RSOS-210320 "A new goniopholidid from the Upper Jurassic Morrison Formation, USA: novel insight into aquatic adaptation toward modern crocodylians" have now received comments from reviewers and would like you to revise the paper in accordance with the reviewer comments and any comments from the Editors. Please note this decision does not guarantee eventual acceptance.

Please submit your revised manuscript and required files (see below) no later than 21 days from today's (ie 08-Jul-2021) date. Note: the ScholarOne system will 'lock' if submission of the revision

is attempted 21 or more days after the deadline. If you do not think you will be able to meet this deadline please contact the editorial office immediately.

on behalf of Professor Peter Haynes (Subject Editor)
openscience@royalsociety.org

Associate Editor Comments to Author:

Thank you for your patience while we secured the reviewer commentary below. Given the comments of reviewers 2 and 3, we recommend that you carefully review and revise the manuscript before resubmitting it for further consideration. Both reviewers note that the presentation of the manuscript can be much improved (and so value added to your work), and we would expect you to take these requests into account. Good luck and we'll look forward to seeing the revision in the near future.

Reviewer comments to Author:

Reviewer: 1
Comments to the Author(s)
Great work on a wonderful specimen.

Reviewer: 2
Comments to the Author(s)
Dear authors,

This is an impressive study and an essential contribution to our understanding of aquatic adaptations in neosuchian crocodyliforms. I placed my comments into the attached pdf, they are minor but given the number of them I recommended "major revision".

Overall, I suggest including a larger figure of the skull of the specimen. The material certainly deserves larger resolution photos and page restrictions no longer apply. Similarly, the authors could illustrate some of their observations derived from the dissecting besides the schematic representation.

I disagree with some details of the conclusions: given the instability of thalattosuchia in phylogenies and the poor sampling of ceratobranchials in the tree makes it difficult to trace the origin of the gular valve closing. Moreover, while the hypothesis of the authors regarding the function of the curved ceratobranchial makes perfect sense, it is a hypothesis not a fact (only a single event in the phylogeny) and the evolutionary implications should be proposed with less certainty.

I am looking forward to see this excellent and essential work to be published!

Reviewer: 3

Comments to the Author(s)

First of all, I'd like to apologise for my review being late. Secondly, I do like the 'thrust' of this manuscript, I just think it requires major modifications to be acceptable for RSOS.

Odd statements

There are some odd comments here and there, such as this from the introduction:

"In the evolutionary history of crocodyliforms, the presence of the secondary palate is a key character for eusuchian crocodyliforms, and it appeared in the successive stem groups of Eusuchia such as paralligatorids and derived goniopholidids "

All mesoeucrocodylians have a maxillopalatine secondary palate, and there is an independent maxillopalatine secondary palate in Shartegosuchoidea (even an independent acquisition of the 'pterygoid secondary palate'). I'm not entirely sure what the authors are meaning here.

Another one is:

"Goniopholidids, which is a transitional group to Eusuchia,"
Goniopholidids aren't. It's an odd thing to say.

And there are more of these. I think the authors need to give a serious re-think to the tone of the manuscript. It reads very teleological, with neosuchian groups being 'pit-stops' on the road to Eusuchia. They weren't, many, such as goniopholidids, were evolutionary successful lineages that persisted for tens of millions of years.

Phylogeny

As the authors will no doubt be aware crocodylomorph phylogeny is chaotic in places. The position of major clades are in flux. So the comment:

"Goniopholididae is a group of basal neosuchian crocodyliforms found in the in the northern hemisphere during the Jurassic and Early Cretaceous, and a sister clade to Paralligatoridae and Eusuchia"

Needs to be better cited. I would simply state that Goniopholididae is a successful lineage of neosuchians outside of the Paralligatoridae + Eusuchia clade. That way, what you are saying doesn't require discussing competing phylogenetic hypotheses.

Also, there is a completely different topology used for the R analyses in the SI... Where does it come from? Why are you not using the topology you analysed with the modified Andrade dataset?

New species

As RSOS is an online-only journal can the authors make sure everything they do is done in accordance with the ICZN Code? That means ensuring at proof stage the ZooBank LSIDs for both the paper and the new species are present.

Anatomy and SI

So, the anteriorly elongated internal nares. I can't see it in the figures. There are no close-ups, only interpretative line drawings. And from those, I would hesitate to say the morphology isn't purely post-mortem damage.

To be honest, the description and figuring are what frustrate me most. As the authors say, this is the most complete known skeleton of a goniopholidid, and the description is largely in the SI and

so are the figures. The complete description needs to be 'main text', there needs to be much more, and better figures.

I'm really sorry, but it's really not acceptable to 'hide away' the vast bulk of the description, and to give such limited figuring. There are some really good figures in the SI, but they are put into multi-element plates that reduce their size and quality (when you zoom in to double check the anatomy). The SI figures should be split into smaller units, and added into the main document.

===PREPARING YOUR MANUSCRIPT===

===PREPARING YOUR REVISION IN SCHOLARONE===

Please ensure that you include a summary of your paper at Step 2 'Type, Title, & Abstract'. This should be no more than 100 words to explain to a non-scientific audience the key findings of your

research. This will be included in a weekly highlights email circulated by the Royal Society press office to national UK, international, and scientific news outlets to promote your work.

Author's Response to Decision Letter for (RSOS-210320.R0)

See Appendix B.

RSOS-210320.R1 (Revision)

Review form: Reviewer 2 (Márton Rabi)

Is the manuscript scientifically sound in its present form?

Yes

Are the interpretations and conclusions justified by the results?

Yes

Is the language acceptable?

No

Do you have any ethical concerns with this paper?

No

Have you any concerns about statistical analyses in this paper?

No

Recommendation?

Accept with minor revision (please list in comments)

Comments to the Author(s)

Dear authors,

Thank you for answering my comments and making the changes. I suggest specifying that the instable phylogenetic position of *Thalattosuchia* is not affecting the conclusions (if that is the case). think the manuscript is otherwise acceptable once minor improvements of the English are made here and there.

Review form: Reviewer 4 (Komsorn Lauprasert)

Is the manuscript scientifically sound in its present form?

Yes

Are the interpretations and conclusions justified by the results?

Yes

Is the language acceptable?

Yes

Do you have any ethical concerns with this paper?

No

Have you any concerns about statistical analyses in this paper?

No

Recommendation?

Accept as is

Comments to the Author(s)

Congratulations, your work is very interesting and valuable for crocodyliforms anatomy, especially the evolution from neosuchia to modern crocodylians.

Decision letter (RSOS-210320.R1)

Dear Dr Yoshida

On behalf of the Editors, we are pleased to inform you that your Manuscript RSOS-210320.R1 "A new goniopholidid from the Upper Jurassic Morrison Formation, USA: novel insight into aquatic adaptation toward modern crocodylians" has been accepted for publication in Royal Society Open Science subject to minor revision in accordance with the referees' reports. Please find the referees' comments along with any feedback from the Editors below my signature.

Please submit your revised manuscript and required files (see below) no later than 7 days from today's (ie 28-Oct-2021) date. Note: the ScholarOne system will 'lock' if submission of the revision is attempted 7 or more days after the deadline. If you do not think you will be able to meet this deadline please contact the editorial office immediately.

on behalf of Peter Haynes (Subject Editor)
openscience@royalsociety.org

Associate Editor Comments to Author:
Comments to the Author:

The reviewers are largely satisfied that your paper is ready for acceptance - congratulations!
There are a couple of (minor) suggestions made by reviewer 2 regarding some phrasing and a

recommendation to check the language once more to help ensure the clarity of your message is maintained, but otherwise, good job on tackling the reviewer feedback and we'll look forward to seeing the paper in the journal after this final revision. All best.

Reviewer comments to Author:

Reviewer: 2

Comments to the Author(s)

Dear authors,

Thank you for answering my comments and making the changes. I suggest specifying that the instable phylogenetic position of *Thalattosuchia* is not affecting the conclusions (if that is the case). think the manuscript is otherwise acceptable once minor improvements of the English are made here and there.

Reviewer: 4

Comments to the Author(s)

Congratulations, your work is very interesting and valuable for crocodyliforms anatomy, especially the evolution from neosuchia to modern crocodylians.

===PREPARING YOUR MANUSCRIPT===

one version should clearly identify all the changes that have been made (for instance, in coloured highlight, in bold text, or tracked changes);

===PREPARING YOUR REVISION IN SCHOLARONE===

-- If you are requesting an article processing charge waiver, you must select the relevant waiver option (if requesting a discretionary waiver, the form should have been uploaded, see 'File upload' above).

-- If you have uploaded any electronic supplementary (ESM) files, please ensure you follow the guidance at <https://royalsociety.org/journals/authors/author-guidelines/#supplementary-material> to include a suitable title and informative caption. An example of appropriate titling and captioning may be found at https://figshare.com/articles/Table_S2_from_Is_there_a_trade-off_between_peak_performance_and_performance_breadth_across_temperatures_for_aerobic_scope_in_teleost_fishes_/3843624.

Author's Response to Decision Letter for (RSOS-210320.R1)

See Appendix C.

Decision letter (RSOS-210320.R2)

Dear Dr Yoshida,

I am pleased to inform you that your manuscript entitled "A new goniopholidid from the Upper Jurassic Morrison Formation, USA: novel insight into aquatic adaptation toward modern crocodylians" is now accepted for publication in Royal Society Open Science.

on behalf of Professor Peter Haynes (Subject Editor)
openscience@royalsociety.org

Appendix A**PROCEEDINGS OF
THE ROYAL SOCIETY B**

BIOLOGICAL SCIENCES

New evidence for a North American origin of modern sea turtles

Journal:	Proceedings B
Manuscript ID	RSPB-2019-1240
Article Type:	Research
Date Submitted by the Author:	28-May-2019
Complete List of Authors:	Gentry, Andrew; University of Alabama at Birmingham, Department of Biology; McWane Science Center, Department of Collections Ebersole, Jun; McWane Science Center, Department of Collections Kiernan, Caitlin; P.O. Box 130593
Subject:	Evolution < BIOLOGY, Palaeontology < BIOLOGY, Taxonomy and Systematics < BIOLOGY
Keywords:	Sea turtles, phylogeny, Demopolis Chalk, fossil calibration, climate change
Proceedings B category:	Evolution

Author-supplied statements

Relevant information will appear here if provided.

Ethics

Does your article include research that required ethical approval or permits?:

This article does not present research with ethical considerations

Statement (if applicable):

CUST_IF_YES_ETHICS :No data available.

Data

It is a condition of publication that data, code and materials supporting your paper are made publicly available. Does your paper present new data?:

Yes

Statement (if applicable):

A detailed geological description of the Demopolis Chalk, locality information for each *Asmodochelys* specimen, a complete list of the characters used in the phylogenetic analysis, the character matrix in NEXUS file format, the molecular constraint tree, and the consensus trees of both the unweighted parsimony and Bayesian analyses are provided as electronic supplementary material.

Conflict of interest

I/We declare we have no competing interests

Statement (if applicable):

CUST_STATE_CONFLICT :No data available.

Authors' contributions

This paper has multiple authors and our individual contributions were as below

Statement (if applicable):

JAE contributed with geological data and analysis. CRK performed fieldwork and provided geological data. ADG designed the research, performed the phylogenetic analysis, and wrote the manuscript / electronic supplementary material with scientific and editorial input from all other authors.

**New evidence for a North American origin of modern sea turtles**

ANDREW D. GENTRY^{1,2*}, JUN A. EBERSOLE², CAITLIN R. KIERNAN³

¹Department of Biology, University of Alabama at Birmingham, Birmingham, Alabama 35233,
USA

²Department of Collections, McWane Science Center, Birmingham, Alabama 35203, USA

³P.O. Box 130593, Mountain Brook, Alabama 35213, USA

*corresponding author

Abstract

Resolving the phylogeny of sea turtles is uniquely challenging given the high potential
for the unification of convergent lineages due to systematic homoplasy. Equivocal
reconstructions of marine turtle evolution subsequently inhibit efforts to establish fossil
calibrations for molecular divergence estimates and prevent the accurate reconciliation of
biogeographic or palaeoclimatic data with phylogenetic hypotheses. Here we describe a new
genus and species of marine turtle, *Asmodochelys parhami*, from the Upper Campanian
Demopolis Chalk of Alabama and Mississippi, USA represented by three partial shells.
Phylogenetic analysis shows that *A. parhami* belongs to the ctenochelyids, an extinct group that
shares characteristics with both pan-chelonioids and pan-cheloniids. In addition to supporting
Ctenochelyidae as a sister taxon of Chelonioidea, our analysis places Protostegidae outside of the
Chelonioidea crown group and recovers *Allopleuron hofmanni* as a stem dermochelyid. Gap
Excess Ratio results indicate a strong stratigraphic congruence of our phylogeny that when
combined with biogeographic data, implies a North American origin of Chelonioidea in the
middle-to-late Campanian, ~20 Myr earlier than current molecular divergence studies suggest.
Placed in the context of palaeoclimatic change, our analysis reveals that rapid shifts in sea
surface temperature may have strongly influenced the historical diversity of marine turtles.

1. Introduction

Recent studies have demonstrated that the incorporation of palaeontological data
improves forecasts of biodiversity responses to climate change [1,2,3]. Refining these predictions
for any particular taxon therefore relies upon a firm understanding of its evolutionary history.
With an extensive fossil record spanning more than 90 Myr, sea turtles (Chelonioidea) are the

oldest living marine tetrapod lineage [4] and with modern species being focal taxa for global
conservation efforts, chelonioids provide a prime model for this type of integrative approach to
biodiversity risk assessment. Unfortunately, the phylogenetic positions of many fossil
chelonioids remain poorly justified, resulting in a lack of definitive fossil calibrations for
molecular divergence estimates [5]. The subsequent uncertainty surrounding the evolution of
chelonioids hinders efforts to produce well resolved phylogenetic hypotheses that can be coupled
with marine geochemical proxies for palaeoclimatic shifts (*e.g.* C, Sr, & O isotope records).

The most problematic group with regard to the composition of total group Chelonioidea
is Protostegidae [6]. Often recovered as highly derived chelonioids [7,8,9], the fossil occurrence
of the oldest protostegid, *Desmatochelys padillai*, predates that of the earliest unambiguous non-
protostegid chelonioid, *Toxochelys latiremis*, by approximately 30 Myr [10,11]. The inclusion of
the protostegids into Chelonioidea also necessitates the existence of a nearly 50 Myr ghost-
lineage for the earliest fossil chelydroid, as molecular evidence strongly supports a sister
relationship between chelonioids and Chelydroidea [12,13]. It has been argued that the recovery
of protostegids as crown group chelonioids in many phylogenetic analyses is the result of
homoplasy due to the inclusion of characters tied to convergent marine specializations in turtle
character-taxon matrices and that protostegids represent an earlier, distinct radiation of marine
adapted turtles [11,14]. More recent studies have indicated that protostegids may be stem
chelonioids [15,16], a scenario that would significantly reduce the implied ghost lineages for the
clades comprising the chelonioid crown group. The true relationship between protostegids and
crown chelonioids can only be resolved through the further refinement of turtle character-taxon
matrices and the inclusion of additional fossil chelonioids into global phylogenetic studies.

Here we describe a new genus and species of fossil chelonioid, *Asmodochelys parhami*,
from the Upper Campanian Demopolis Chalk (79-74.5 Ma [17]) of the Gulf Coastal Plain, USA.
This new taxon is included in an expanded phylogenetic analysis of turtles which indicates that
*Asmodochelys* belongs to the extinct Ctenochelyidae, a pan-chelonioid group characterized by a
laterally serrated shell, extensive costal and plastral fontanelles, and the presence of epineurals
positioned at various intervals along the neural series. Our analysis also recovers a novel
phylogeny for marine turtles that, when combined with stratigraphic and biogeographic
evidence, suggests a North American origin of crown group Chelonioidea in the middle to late
Campanian.

2. Materials and methods

Three specimens of the new stem chelonioid are known, all from the Upper Campanian
Demopolis Chalk of Alabama and Mississippi, USA. Bayesian and parsimony phylogenetic
analysis were used to establish the phylogenetic position of the new taxon. Our matrix was
constructed using a modified version of the Evers & Benson [9] character-taxon matrix which
greatly expanded on previous matrices [11,18,19,20] and is the most comprehensive character
taxon matrix for turtles to date (see electronic supplementary material for complete character
list). The scorings and character definitions generally follow those of Evers & Benson [9] with
the following exceptions:

1. The addition of six fossil chelonioid species: *Toxochelys latiremis*, *Ctenochelys stenoporus*,
*Ctenochelys acris*, *Prionocheilus matutina*, *Peritresius ornatus*, and *Asmodochelys parhami* (see
electronic supplementary material for sources of character scoring; table S1)

2. The creation of two new characters: ch. 203: presence of epineurals; ch. 309: maximum width
of coracoid posterior process.

3. Nine revised character definitions: ch. 55, ch. 65, ch. 93, ch. 103, ch. 119, ch. 212, ch. 213, ch.
314, ch. 325.

4. We rescore three characters for *Allopleuron hofmanni* (ch. 211, ch. 218, ch. 314), one
character for *Protostega gigas* (ch. 211), three characters for *Lepidochelys olivacea* (ch. 182, ch.
201, ch. 202), one character for *Caretta caretta* (ch. 314), and two characters for *Chelonia*
*mydas*, *Lepidochelys kempii*, and *Natator depressus* (ch. 201, ch. 202).

These modifications resulted in a matrix of 87 species and 347 characters (electronic
supplementary material, data S1). The matrix was edited using Mesquite v.3.6 build 917 [21]. Of
the 347 total characters, 24 multistate characters were treated as ordered. Parsimony analyses
were conducted in PAUP* v.4.0a build 165 [22] using the heuristic search algorithm with 10,000
random addition sequence replicates of starting trees obtained by simple stepwise addition and
the tree bisection and reconnection method of branch swapping. Characters scored as multiple
states for any species were treated as polymorphisms and branches with a maximum length = 0
were set to collapse. Bremer decay index (BDI) values were calculated manually by retaining
trees with sequentially higher step values than the most parsimonious trees until all but the most
well-supported bipartitions ($BDI \geq 6$) had collapsed. Bayesian analysis was conducted in
MrBayes v.3.2 [23] using a general-time reversible substitution type and *invgamma* model of
rate variation. Model parameters, posterior distribution and branch lengths were estimated with
Markov chain Monte Carlo, using four chains of 10,000,000 generations with sampling every
1,000 generations. Analyses were run until the average standard deviation of the split frequencies
was below 0.01. The first 25% of samples were discarded as burn-in. In both the parsimony and

Bayesian analyses, *Proganochelys quenstedti* was set as the outgroup and the positions of extant
species were constrained using a molecular ‘backbone’ (figure S16) derived from a global
phylogenomic analysis of turtles [13]. To reduce the potential for chimeric operational
taxonomic units (OTUs), supraspecific OTUs were excluded. Phylogenetic definitions follow
Joyce et al. [24].

3. Systematic palaeontology

Testudines [25]

Cryptodira [26]

Pan-Chelonioidea [24]

Ctenochelyidae [27]

*Asmodochelys parhami* n. gen. et sp.

urn:lsid:zoobank.org:act: 147C2B3C-F3A2-4818-879E-452ADE2C4DE3

(a) Etymology

*Asmodo* from the Greek ‘Asmodaios’, the horned deity and Master of the Sea who, according to
Islamic legend, was entombed in stone on the ocean floor [28] and *chelys* from the Greek
‘kelyfos’ meaning shell. The species name honours James F. Parham, former Curator of
Palaeontology at the Alabama Museum of Natural History for his contributions to Alabama
palaeontology and the study of marine turtle evolution.

(b) Holotype

MSC (McWane Science Center, Birmingham, AL) 35984. A single individual preserving the
nuchal, four neurals, two epineurals, eight left peripherals, five right peripherals, a partial first
suprapygal, two costals of the left side, two costals of the right side, approximately half of the
left hyoplastron, and one cervical vertebra (figure 1).

(c) Type locality and horizon

Town of Alberta, Wilcox County, AL, USA. “Muldrow” Member of the Demopolis Chalk
Formation, Upper Campanian (see electronic supplementary material for detailed locality
information).

(d) Paratypes

MMNS (Mississippi Museum of Natural Sciences, Jackson, MS) 3958, site MS.53.017,
Oktibbeha County, Mississippi, USA, “Muldrow” Member of the Demopolis Chalk, Upper
Campanian. This specimen preserves two left peripherals, one neural, one epineural, both
suprapygals, and a complete pygal (figure 1). MSC 40935, site ASu-14, Sumter County,
Alabama, USA, Bluffport Marl Member of the Demopolis Chalk, Upper Campanian. This
specimen consists of one complete costal, three medial peripherals, four posterior peripherals,
three neurals, two epineurals, the first suprapygal, and the left xiphiplastron (figure S7; see
electronic supplementary material for additional locality and specimen information).

(e) Diagnosis

Thick shell with a deep nuchal embayment; nuchal fontanelles absent; horn-like protuberance on
the anterodorsal edge of the first peripheral; concave dorsal plates of peripherals 4-8 resulting in

the formation of a pronounced peripheral gutter; extreme reduction in height and width of
posterior peripherals; anterior and posterior neurals wider than long; four neural keel elevations
with epineurals dorsal to the junctions of neurals 1-2, 2-3, 4-5, 6-7; single keel elevation dorsal
to the first suprapygal terminating immediately anterior to the second suprapygal; dorsal facet of
the pygal considerably longer than the ventral facet; distinct notch at the posterior margin of the
pygal.

(f) Comparative diagnosis

*Asmodochelys parhami* can be distinguished from all previously described ctenochelyid turtles
by the following carapacial characteristics: (i) The nuchal embayment of *Ctenochelys* and
*Peritresius* receives only minimal contributions from the medial margins of the left and right first
peripheral, whereas more than half of the nuchal embayment of *Asmodochelys* is formed by the
first peripherals. (ii) Nuchal fontanelles are present in *Ctenochelys*, *Prionocheles*, and *Peritresius*
whereas these features are absent in *Asmodochelys*. (iii) The lateral peripherals of *Asmodochelys*
are widest at the level of the suture between the second and third costal plate differing from the
condition observed in *Ctenochelys* and *Prionocheles* where the peripherals are widest along the
posterior edge of the carapace. (iv) The epineural dorsal to the contact between the first and
second neural of *Asmodochelys* is absent in *Ctenochelys*, *Prionocheles*, and *Peritresius*. (v)
Additional characters from the diagnosis are unknown in *Ctenochelys*, *Prionocheles*, and
*Peritresius* such as the extreme reduction in the size of the posterior peripherals, the horn-like
protuberance on the anterodorsal edge of the first peripheral, and the varying length of the dorsal
and ventral facets of the pygal.

4. Description

The carapace of *Asmodochelys* is strongly cordiform and is much longer than wide (maximum
carapace length = ~1.0-1.5m) with the widest point being at the level of the fifth peripheral
(figure 1). The nuchal bears a pronounced medial embayment which extends laterally
approximately half of the total width of the nuchal and receives significant contributions from
the anteromedial edge of both the left and right first peripherals (figure 1a). Due to the posterior
convexity of the nuchal, the suture between the nuchal and first peripheral lies at a 100-110°
angle with the sagittal midline of the carapace, differing from other known ctenochelyids,
including *Ctenochelys* [29], *Prionocheles* [10,27], and *Peritresius* [30]. A raised pedestal
preserved on the visceral surface of the nuchal likely served as an articulation site for the
transverse process of the eighth cervical vertebra, a trait proposed as an apomorphy of pan-
chelonoids [14,31]. The nuchal bears a slight ridge running along the dorsal midline beginning
immediately posterior to the posteromedial edge of the cervical scute which increases in camber
as it progresses posteriorly towards the sutural articulation with the first neural. The costo-nuchal
sutures span the entirety of the posterolateral margins of the nuchal and extend anterolaterally,
terminating immediately posterior to the contact between the nuchal and first peripheral. Based
on the extent of the costo-nuchal sutures and morphology of the articulation between the nuchal
and first neural, there is no indication that nuchal fontanelles were present, differing from the
condition seen in other ctenochelyid marine turtles (e.g. *Ctenochelys*, *Prionocheles*, and
*Peritresius*) where these fontanelles are found in all ontogenetic stages [27]. The cervical scute
of *Asmodochelys* roughly resembles that of *Ctenochelys stenoporus* [29], forming an irregular
heptagonal polygon. However, the cervical scute of *Asmodochelys* is proportionally longer and
covers a much larger percentage of the dorsal surface of the nuchal. The anterior margin of the

first peripheral bears a dorsally oriented, horn-like protuberance (a distinctive character of
*Asmodochelys*) that forms the lateral-most extent of the nuchal embayment (figure 1a). The
presence of this feature is unknown among the Late Cretaceous chelonioids of North America
but has been noted to a lesser extent in *Allopleuron hofmanni* from the Maastrichtian of Europe
[32]. The dorsolateral and ventral surfaces of the medial peripherals are widely separated (figure
1h-o,r,s), forming a high, proximally facing sulcus which runs from the posterior half of the
third peripheral to the anterior half of the eighth, resembling the condition observed in
*Peritresius* [30]. Dorsally, the anteromedial peripherals bear a ventrally convex trough which
terminates on the dorsal surface of the eighth peripheral. The height and width of the posterior
peripherals are greatly reduced (figure 1y,z), similar to those of *Allopleuron hofmanni* [32]. The
reduction in width of the posterior peripherals distinguishes *Asmodochelys* from other
ctenochelyids such as *Ctenochelys* and *Prionocheles* whose peripherals widen posteriorly along
the series and continue to increase in width during ontogeny. Based on the estimated size of the
carapace and the presence of laterally expanded costal plates (figure 1c,g,t,w) in both of the most
complete specimens of *Asmodochelys* (MSC 35984 and MSC 40935), it is likely that these
specimens represent mature individuals and that the relative proportions of the peripheral series
would not differ significantly in a more ontogenetically advanced individual.

The neural series of *Asmodochelys* is comprised of nine neurals and four epineurals
(figure 1d-f,p,q,u). The morphology of the generally hexagonal, dorsally keeled neurals of
*Asmodochelys* resembles that of the pan-chelonioids *Ctenochelys* and *Prionocheles* but differs in
that the width of each neural often equals or exceeds its length, similar to those of *Peritresius*
*ornatus* [30]. Vertebral scale sulci are visible on the dorsal surface of neurals two and six (figure
1f,p). The neurals of *Asmodochelys* lack the distinctive dermal sculpturing of *Peritresius ornatus*

although the external surface is marked by numerous vascular innervations, though somewhat
less prominent than those observed in both modern cheloniids [33] and on the costal plates of an
unnamed Oligocene pan-cheloniid [34]. In ventral aspect, the neurals of *Asmodochelys* possess
an extensive layer of notably osteoporotic trabecular bone. The epineurals dorsal to the neural
series form four distinct elevations along the midsagittal keel of the carapace (figure 1),
somewhat similar to the epineurals of *Ctenochelys* and *Prionocheles*. However, the presence of
an epineural between the first and second neural distinguishes *Asmodochelys* from other
ctenochelyids.

The first suprapygal is roughly triangular, tapering in width posteriorly (figure 1v,x),
resembling the first suprapygal of *Peritresius*. The first suprapygal of MMNS 3958 bears a
dorsally rounded keel elevation possibly comprised of one or more episuprapygals (a feature
observed in other ctenochelyids) but due to poor preservation, this arrangement cannot be
determined with any confidence. The second suprapygal is much narrower than the first and
contacts the pygal posteriorly along a broadly concave transverse suture (figure 1x). The dorsal
plate of the pygal is remarkably long (approx. 1.5 times the length of the ventral plate; figure
S5), differing from the equally long ventral and dorsal pygal surfaces of the Santonian-
Campanian pan-chelonioids (e.g. *Toxochelys*, *Ctenochelys*) and the equally short pygal surfaces
observed in the predominantly Maastrichtian pan-chelonioids such as *Allopleuron* and
*Peritresius*. Overall, the carapacial elements of *Asmodochelys* are remarkably robust in their
general construction owing primarily to a 3-5 mm thick layer of dense external cortical bone
(figure S8). The reduction of the compact external cortex and the homogenization of cortical and
interior trabecular bone found in *Ctenochelys* and *Toxochelys* [33] is absent in *Asmodochelys*.
The compact, well vascularized external cortex of *Asmodochelys* more closely resembles the

condition seen in *Allopleuron hofmanni* and may be indicative of a near-shore marine ecology
[33,35]. One procoelous cervical vertebra is preserved with MSC 35984 (figure S6) bearing a
pronounced longitudinal keel along the ventral surface of the centrum, a previously proposed
synapomorphy of pan-chelonioids [11].

5. Phylogenetic analysis

The phylogenetic position of *Asmodochelys* was tested with both parsimony and
Bayesian phylogenetic inference. The unweighted parsimony analysis retrieved 252 most
parsimonious trees (MPTs) with a length of 1592 steps, consistency index of 0.29, retention
index of 0.66, and homoplasy index of 0.75. The strict consensus of these MPTs places
*Asmodochelys* as a basal member of Ctenochelyidae with *Ctenochelys*, *Prionocheilus*, and
*Peritresius*, together forming a sister clade to Chelonioidea (figure S17). Ctenochelyidae is
supported by five unambiguous synapomorphies: (1) a moderate contribution to the upper
triturating surface by the palatine (ch. 55); (2) shallow ridge on the ventral surface of the vomer
(ch. 65); (3) a domed shape contribution of the vomer to palate roof (ch. 66); (4) the presence of
epineurals (ch. 203); and (5) the lateral process of the humerus being slightly separated from the
caput humeri (ch. 325). Certain ctenochelyids have historically been recovered as sister taxa to
*Toxochelys* on either the stem of Chelonioidea [8,36] or within Pan-Cheloniidae
[7,32,37,38,39,40]. *Toxochelys* was recovered here as a stem chelonioid and sister taxon to the
clade formed by Ctenochelyidae and crown group Chelonioidea. In contrast with many previous
analyses [8,9,11,37,41,42], Protostegidae is recovered as a clade of stem chelonioids supported
by five unambiguous synapomorphies: (1) presence of a medial contact of the palatines (ch. 62);
(2) pterygoids contact the medial edge of the mandibular condyle facet (ch. 103); (3) strongly

serrated lateral and medial margins of the plastron (ch. 237); (4) expansion of the lateral process
onto the ventral surface of the humerus (ch. 330); (5) lateral process of the humerus with
prominent anterior projection (ch. 331). Our analysis also recovers *Allopleuron* as a basal
member of Pan-Dermochelyidae, the sister clade to Pan-Cheloniidae, that together form
Chelonioidea. Chelonioidea is supported by six unambiguous synapomorphies: (1) contact
between the parietal and squamosal (ch. 15); (2) absence of a parasagittal ridge on the palatal
surface of the pterygoid (ch. 104); (3) rod-like rostrum basisphenoidale (ch. 138); (4) vertical
median ridge on the anterior surface of the dorsum sellae (ch. 140); and (5) humerus with a V-
shaped lateral process (ch. 329).

A growing body of evidence is available which suggests that implementing mild implied
weighting improves the results of cladistic analyses using parsimony [43,44]. To test the
influence of implied weighting, a second tree search was conducted using mild weighting with a
*k* factor of 12. This analysis retrieved tree topologies very similar to those obtained in the
unweighted analysis, with the only exceptions being the removal of the macrobaenid *Judithemys*
from the crown of Cryptodira and *Solnhofia* being recovered as a member of Thalassocheyleia
(figure 2a). As we use only a mild weighting and as the placement of these taxa in the weighted
analysis is more consistent with previous cladistic studies of Thalassocheyleia [19,45] and
Macrobaenidae [46], we consider the weighted topology to be preferable. Branch support of
clades, evaluated using Bremer support, retrieved strong support for most of the nodes inside
Pan-Chelonioidea. Bremer values higher than one were recovered for Angolachelonia (3),
Protostegidae (3), the clade formed by *Toxochelys* and more derived pan-chelonioids (5), Pan-
Dermochelyidae (3), and Chelonioidea (5).

Overall, the tree topologies recovered by the Bayesian analyses were similar to those of
the parsimony analyses, with Protostegidae outside of crown group Chelonioidea and
Angolachelonia sister to a clade consisting of *Toxochelys latiremis* and the more crownward
members of Pan-Chelonioidea in the 50% majority-rule consensus tree (electronic supplementary
material, figure S18). However, the relationships within Pan-Chelonioidea are somewhat less
resolved with almost all Late Cretaceous stem chelonioids and the Cenozoic stem cheloniids
*Puppigerus*, *Eochelone*, and *Argillochelys* forming a large polytomy. *Peritresius martini* is
recovered as a sister taxon to *Allopleuron hofmanni* on the stem of Dermochelyidae and the clade
formed by *Archelon* and *Protostega* is separated from the other members of Protostegidae.
Support values are relatively high, with more than 60% support for almost all resolved nodes.

Gap Excess Ratio (GER) values [47] were calculated to assess the stratigraphic
congruence of our preferred topology (figure 2b) relative to that of the prevailing historical
marine turtle phylogenetic hypothesis (figure 2c). Our analysis recovered a significantly higher
GER score despite the placement of the Late Jurassic thalassochelydians within Pan-
Chelonioidea and the resulting increase in the implied ghost lineages for Chelydroidea and
Protostegidae. This result is due to the exclusion of Protostegidae from crown-group
Chelonioidea and the consequent reduction in the implied ghost lineages for both Pan-
Cheloniidae and Pan-Dermochelyidae. The position of *Allopleuron* in our analysis further
reduces the implied ghost lineage for Pan-Dermochelyidae by replacing *Eosphargis* as the
earliest stem dermochelyid.

6. Discussion

Our phylogenetic analyses provide strong support for the placement of angolachelonians,
protostegids, and ctenochelyids as stem chelonioids. The placement of protostegids as stem
chelonioids supports the conclusions of the most recent phylogenetic analyses of chelonioids
[15,16] and is more congruent with the fossil record than historical hypotheses of marine turtle
evolution while still supporting the proposed singular origin of a pelagic ecology among non-
pleurodiran turtles [9]. Biogeographic evidence indicates that although the earliest protostegids
were primarily restricted to the Southern Hemisphere, by the Late Cretaceous the protostegids
had become a predominantly North American lineage (figure 3). This transition is temporally
coincident with the opening of the Equatorial Atlantic Gateway during the Turonian [48]. As
almost all Late Cretaceous non-protostegid chelonioids are also North American, the exclusion
of protostegids from crown group Chelonioidea resolves the biogeographic issues associated
with the placement of protostegids as derived dermochelyoids. Furthermore, the diversification
of non-protostegid sea turtles during the Campanian took place following the extinction of most
species of protostegid [37]. This scenario supports the previously hypothesized pattern of
ecological replacement following the extinction of similarly adapted forms within the Pan-
Chelonioidea lineage [49]. Our analyses indicate that the youngest stem chelonioids and the
oldest stem cheloniids are likely North American taxa, supporting the biogeographic origin of
modern chelonioids proposed by both fossil [5,11] and molecular [12] studies of turtles. The
sister taxa relationship between *Allopleuron* and *Peritresius martini* suggested by our Bayesian
analysis means that we cannot rule out the possibility that certain Late Cretaceous ctenochelyids
from North America may be early stem dermochelyids.

When the phylogeny presented here is coupled with oceanic palaeotemperature estimates
inferred from foraminiferal oxygen isotope measurements [50], an interesting pattern emerges.

The Cretaceous and Paleogene phylogenetic diversification events of the major pan-chelonioid
lineages occur during periods of extreme warmth compared to modern day (figure 3). A
diversification of protostegids during the early Albian immediately proceeded the Aptian/Albian
Boundary Interval (AABI). The largest radiation of non-protostegid stem chelonioids occurred
during the late Santonian/early Campanian following the Cretaceous Thermal Maximum (KTM)
and a late Paleocene/early Eocene diversification of crown chelonioids occurred in close
temporal proximity to the Paleocene-Eocene Thermal Maximum. The diversity of stem
chelonioids appears to have been highest in the middle-late Cretaceous towards the end of an
approximately 11.5 Myr climate interval known as the Cretaceous Hot Greenhouse [50] when
sea surface temperatures are thought to have been 15-25°C higher than present day. However,
chelonioid diversity decreases during peaks in oceanic temperatures (e.g. AABI & KTM),
possibly a result of declines in marine primary productivity due to Oceanic Anoxic Events
(OAEs) known to have occurred concomitantly with these thermal maxima [51,52]. Chelonioid
diversity also dramatically declines during periods of rapid global cooling, as implied by the
observed reduction in species across the Campanian/Maastrichtian boundary [53]. This evidence
suggests that although marine turtles appear to have thrived during past periods of extreme
warmth, they are clearly sensitive to rapid changes in temperature, regardless of the nature of
those changes.

Acknowledgements

We thank Sandy Ebersole of the Alabama Geological Survey for the shape files necessary to
make figure S1, George Phillips for allowing access to the specimens in his care, and Serjoscha
Evers for sharing photographs of important fossil material used in the creation of the character

matrix. Silhouettes used in figure S16 created by Roberto Díaz Sibaja, Neil Kelley, and Andrew
Farke are distributed by PhyloPic under CCY License 3.0.

References

- 1. Blois JL, Zarnetske PL, Fitzpatrick MC, Finnegan S. 2013 Climate change and the past,
present, and future of biotic interactions. *Science* **341**, 499–504.
(doi:10.1126/science.1237184)
- 2. Maguire KC, Nieto-Lugilde D, Fitzpatrick MC, Williams JW, Blois JL. 2015 Modeling
species and community responses to past, present, and future episodes of climatic and
ecological change. *Annu. Rev. Ecol. Evol. Syst.* **46**, 343 – 368. (doi:10.1146/annurev-
ecolsys-112414-054441)
- 3. Jones LA, Mannion PD, Farnsworth A, Valdes PJ, Kelland S-J, Allison PA. 2019
Coupling of palaeontological and neontological reef coral data improves forecasts of
biodiversity responses under global climate change. *R. Soc. open sci.* **6**, 18211.
(doi:10.1098/rsos.182111)
- 4. Pyenson ND, Kelley NP, Parham JF. 2014 Marine tetrapod macroevolution: Physical and
biological drivers on 250 Ma of invasions and evolution in ocean ecosystems. *Palaeog.*,
*Palaeocl.*, *Palaeoec.* **400**, 1 – 8. (doi:10.1016/j.palaeo.2014.02.018)
- 5. Joyce WG, Parham JF, Lyson TR, Warnock RC, Donoghue PC. 2013 A divergence
dating analysis of turtles using fossil calibrations: an example of best practices. *J.*
*Paleontol.* **87**, 612 – 634. (doi:10.1666/12-149)
- 6. Zangerl R. 1953a The vertebrate fauna of the Selma Formation of Alabama, part III: the
turtles of the family Protostegidae. *Field. Geol. Mem.* **3**, 1 – 136.

- 7. Hirayama R. 1997 Distribution and diversity of Cretaceous chelonoids. In *Ancient*
*marine reptiles* (eds. J Callaways, E Nicholls), pp. 225 – 241. Elsevier.
- 8. Kear BP, Lee MS. 2006 A primitive protostegid from Australia and early sea turtle
evolution. *Bio. Letters* **2**, 116 – 119. (doi:10.1098/rsbl.2005.0406)
- 9. Evers SW, Benson RJ. 2018 A new phylogenetic hypothesis of turtles with implications
for the timing and number of evolutionary transitions to marine lifestyles in the group.
*Palaeontology*. **62**, 93 – 134. (doi:10.1111/pala.12384)
- 10. Zangerl R. 1953b The vertebrate fauna of the Selma Formation of Alabama, part IV: the
turtles of the family Toxochelyidae. *Field. Geol. Mem.* **3**, 137 – 277.
- 11. Cadena EA, Parham JF. 2015 Oldest known marine turtle? A new protostegid from the
Lower Cretaceous of Colombia. *PaleoBios* **32**, 1 – 42.
- 12. Crawford NG, Parham JF, Sellas AB, Faircloth BC, Glenn TC, Papenfuss TJ, Henderson
JB, Hansen MH, Simison WB. 2015 A phylogenomic analysis of turtles. *Mol.*
*Phylogenet. Evol.* **83**, 250 – 257. (dx.doi:10.1016/j.ympcv.2014.10.021)
- 13. Pereira A, Sterli J, Moreira F, Schrago C. 2017 Multilocus phylogeny and statistical
biogeography clarify the evolutionary history of major lineages of turtles. *Mol.*
*Phylogenet. Evol.* **113**, 59 – 66. (dx.doi:10.1016/j.ympcv.2017.05.008)
- 14. Joyce WG. 2007 Phylogenetic relationships of Mesozoic turtles. *Bull. Peab. Mus. Nat.*
*Hist.* **48**, 3 – 102.
- 15. Raselli I. 2018 Comparative cranial morphology of the Late Cretaceous protostegid
*Desmatochelys lowii*. *PeerJ Preprints* **6**, e26863v1.

- 16. Evers SW, Barrett PM, Benson RB. 2019 Anatomy of *Rhinochelys pulchriceps*
(Protostegidae) and marine adaptation during the early evolution of chelonioids. *PeerJ* **7**,
e6811. (doi:10.7717/peerj.6811)
- 17. Puckett TM. 2005 Santonian-Maastrichtian planktonic foraminiferal and ostracode
biostratigraphy of the northern Gulf Coastal Plain, USA. *Stratigraphy* **2**, 117 – 146.
- 18. Cadena EA. 2015 The first South American sandownid turtle from the Lower Cretaceous
of Colombia. *PeerJ* **3**, e1431. (doi:10.7717/peerj.1431)
- 19. Anquetin J, Püntener C. 2015 *Portlandemys gracilis* n. sp., a new coastal marine turtle
from the Late Jurassic of Porrentruy (Switzerland) and a reconsideration of plesiochelyid
cranial anatomy. *PLoS One* **10**, e0129193. (doi:10.1371/journal.pone.0129193)
- 20. Zhou C-F, Rabi M. 2015 A sinemydid turtle from the Jehol Biota provides insights into
the basal divergence of crown turtles. *Sci. Rep.* **5**, 16299. (doi:10.1038/srep16299)
- 21. Maddison WP, Maddison DR. 2019 Mesquite: a modular system for evolutionary
analysis. Version 3.6. (<http://mesquiteproject.org>)
- 22. Swofford DL. 2002 *PAUP**. *Phylogenetic analysis using parsimony (*and other*
*methods)*. Version 4. Sunderland, MA: Sinauer Associates.
- 23. Ronquist F *et al.* 2012 MrBayes 3.2: efficient Bayesian phylogenetic inference and model
choice across a large model space. *Syst. Biol.* **61**, 539 – 542. (doi:10.1093/sysbio/sys029)
- 24. Joyce WG, Parham JF, Gauthier J-A. 2004 Developing a protocol for the conversion of
rank-based taxon names to phylogenetically defined clade names, as exemplified by
turtles. *J. Paleontol.* **78**, 989 – 1013. (doi:10.1666/0022)
- 25. Batsch A. 1788 *Versuch einer Anleitung, zur Kenntniß und Geschichte der Thiere und*
*Mineralien*. Akademische Buchhandlung, Jena.

- 26. Cope ED. 1868 On the origin of genera. *Proc. Acad. Nat. Sci. Phil.* **20**, 242 – 300.
- 27. Gentry A. 2018 *Prionochelys matutina* Zangerl, 1953 (Testudines: Pan-Cheloniidae)
- from the Late Cretaceous of the United States and the evolution of epithecal ossifications
- in marine turtles. *PeerJ* **6**, e5876. (doi:10.7717/peerj5876)
- 28. Klar M. 2004 And we cast upon his throne a mere body: A historiographical reading of
- Q. 38:34. *J. of Qur'anic Studies* **6**, pp. 103 – 126.
- 29. Matzke A. 2007 An almost complete juvenile specimen of the cheloniid turtle
- *Ctenochelys stenoporus* (Hay, 1905) from the Upper Cretaceous Niobrara Formation of
- Kansas, USA. *Palaeontology* **50**, 669 – 691.
- 30. Baird D. 1964 A fossil sea-turtle from New Jersey. *New Jer. St. Mus. Inv.* **1**, 3 – 26.
- 31. Anquetin J. 2012 Reassessment of the phylogenetic interrelationships of basal turtles
- (Testudinata). *J. Vertebr. Paleontol.* **28**, 123 – 133.
- 32. Mulder E. 2003 Comparative osteology, palaeoecology, and systematics of the Late
- Cretaceous turtle *Allopleuron hofmanni* (Gray, 1831) from the Maastrichtian type area. In
- *On the Late Cretaceous tetrapods from the Maastrichtian type area* (eds. E Mulder), pp.
- 23 – 92. Publicaties van het Natuurhistorich Genootschap in Limburg.
- 33. Scheyer TM, Danilov IG, Sukhanov VB, Syromyatnikova EV. 2014 The shell bone
- histology of fossil and extant marine turtles revisited. *Biol. J. Linn. Soc.* **112**, 701 – 718.
- (doi:10.1111/bij.12265)
- 34. Cadena EA, Abella J, Gregori M. 2018 The first Oligocene sea turtle (Pan-Cheloniidae)
- record of South America. *PeerJ* **6**, e4554. (doi:10.7717/peerj.4554)
- 35. Scheyer TM, Sander PM. 2007 Shell bone histology indicates terrestrial palaeoecology of
- basal turtles. *Proc. R. Soc. B* **274**, 1885 – 1893. (doi:10.1098/rspb.2007.0499)

- 36. Gaffney ES, Meylan PA. 1988 A phylogeny of turtles. In *The phylogeny and*
*classification of the tetrapods*. In *Amphibians, Reptiles, Birds* (ed. MJ Benton),
Systematics Association Special Volume, **35**, pp. 157 – 219. Clarendon Press, Oxford.
- 37. Hirayama R. 1994 Phylogenetic systematic of chelonioid sea turtles. *Island Arc* **3**, 270 –
284.
- 38. Hirayama R. 1998 Oldest known sea turtle. *Nature* **392**, 705 – 708.
- 39. Parham JF, Fastovsky DE. 1997 The phylogeny of cheloniid sea turtles revisited.
*Chelonian Conserv. Biol.* **4**, 548 – 554.
- 40. Lehman TM, Thomlinson SL. 2004 *Terlinguachelys fischbecki*, a new genus and species
of sea turtle (Chelonioidea: Protostegidae) from the Upper Cretaceous of Texas. *J.*
*Paleontol.* **78**, 1163 – 1178. (doi:10.1666/0022-
3360(2004)078<1163:TFANGA>2.0.CO;2)
- 41. Tong H, Meylan PA. 2013 Morphology and relationships of *Brachyopsemys tingitana*
gen. et sp. nov. from the Early Paleocene of Morocco and recognition of the new
eucryptodiran turtle family: Sandownidae. In *Morphology and Evolution of Turtles,*
*Vertebrate Paleobiology and Paleoanthropology* (eds. D Brinkman, P Holroyd, J
Gardner), pp. 187 – 212. Dordrecht: Springer. (doi: 10.1007/978-94-007-4309-0_13)
- 42. Bardet N, Jalil N-E, Lapparent de Broin F, Germain D, Lamber O, Amaghazaz M. 2013 A
giant chelonioid turtle from the Late Cretaceous of Morocco with a suction feeding
apparatus unique among tetrapods. *PLoS One* **8**, e63586.
(doi:10.1371/journal.pone.0063586)

- 43. Goloboff PA, Torres A, and Arias JS. 2017 Weighted parsimony outperforms other
methods of phylogenetic inference under models appropriate for morphology. *Cladistics*
**34**, 407 – 437. (doi:10.1111/cla.12205)
- 44. Joyce WG, Lyson TR. 2017 The shell morphology of the latest Cretaceous
(Maastrichtian) trionychid turtle *Helopanoplia distincta*. *PeerJ* **5**, e4169.
(doi:10.7717/peerj.4169)
- 45. Anquetin J, Püntener C, Joyce WG 2017 A review of the fossil record turtles of clade
*Thalassocheilydia*. *Bull. Pea. Mus. Nat. Hist.* **58**, 317 – 369. (doi:10.3374/014.058.0205)
- 46. Parham JF, Hutchinson JH. 2003 A new eucryptodiran turtle from the Late Cretaceous of
North America (Dinosaur Provincial Park, Alberta, Canada). *J. Vertebr. Paleontol.* **23**,
783 – 798.
- 47. Wills MA. 1999 Congruence between phylogeny and stratigraphy: randomization tests
and the gap excess ratio. *Syst. Biol.* **48**, 559 – 580.
- 48. Moullade M *et al.* 1998 Mesozoic biostratigraphic, paleoenvironmental, and
paleobiogeographic synthesis, equatorial Atlantic. In *Proceedings of the Ocean Drilling*
*Program: Scientific Results* (eds. J Mascle, G Lohmann, M Moullade), pp. 481 – 490.
- 49. Parham JF, Pyenson ND. 2010 New sea turtle from the Miocene of Peru and the iterative
evolution of feeding ecomorphologies since the Cretaceous. *J. Paleontol.* **84**, 231 – 247.
- 50. Huber BT, MacLeod KG, Watkins DK, Coffin MF. 2018 The rise and fall of the
Cretaceous Hot Greenhouse climate. *Global and Planetary Change* **167**, 1 – 23.
(doi:10.1016/j.gloplacha.2018.04.004)
- 51. Jenkyns HC, Dickson AJ, Ruhl M, van den Boorn S, 2017 Basalt–seawater interaction,
the Plenus Cold Event, enhanced weathering and geochemical change: deconstructing

- Oceanic Anoxic Event 2 (Cenomanian–Turonian, Late Cretaceous). *Sedimentology* **64**,
16 – 43. (doi:10.1111/sed.12305)
- 52. Huber BT, MacLeod KG, Gröcke D, Kucera M. 2011 Paleotemperature and paleosalinity
inferences and chemostratigraphy across the Aptian/Albian boundary in the
subtropical North Atlantic. *Paleoceanography* **26**, PA4221.
(doi:10.1029/2011PA002178)
- 53. Gentry A, Parham F, Ehret D, Ebersole J. 2018 A new species of *Peritresius* Leidy, 1865
(Testudines: Pan-Cheloniidae) from the Late Cretaceous (Campanian) of Alabama, USA
and the occurrence of the genus within the Mississippi Embayment of North America.
*PLoS One* **13**, e0196551. (doi:10.1371/journal.pone.0195651)

Representative elements and reconstruction of *Asmodochelys parhami*. (a) Nuchal, first and second left peripherals, and first right peripheral (MSC 35984) in dorsal view. (b) Third right peripheral (MSC 35984) in dorsal view. (c) Second right costal (MSC 35984) in dorsal view. (d,e) Fourth neural and second epineural (MSC 35984) in dorsal (d) and left lateral (e) views. (f) First and second neural (MSC 35984) in dorsal view. (g) Fourth right costal (MSC 35984) in dorsal view. (h,i) Fifth right peripheral in posterior (h) and dorsal (i) views. (j,k) Fourth left peripheral (MSC 35984) in dorsal (j) and posterior (k) views. (l,m) Seventh right peripheral (MSC 35984) in posterior (l) and dorsal (m) views. (n,o) Fifth left peripheral (MSC 35984) in dorsal (n) and posterior (o) views. (p,q) Sixth neural and third epineural (MSC 35984) in dorsal (p) and right lateral (q) views. (r,s) Sixth left peripheral (MSC 35984) in dorsal (r) and posterior (s) views. (t) Fifth left costal (MSC 35984) in dorsal view. (u) Seventh neural (MSC 35984) in dorsal view. (v) First suprapygals and pygal (MSC 35984) in dorsal view. (w) Sixth right costal (MSC 35984) in dorsal view. (x) Suprapygals and pygal (MMNS 3958) in dorsal view. (y) Tenth left peripheral (MMNS 3958) in posterior view. (z) Eighth – tenth right peripherals (MMNS 3958) in dorsal view. Dashed lines represent scute sulci.

165x222mm (300 x 300 DPI)

Phylogenetic relationships of *Asmodochelys parhami* and stratigraphic congruence analyses. (a) Strict consensus tree from weighted parsimony analysis. Bremer support values are shown for each resolved node. Dashes indicate Bremer support values ≥ 6 . MPTs = most parsimonious trees. (b) Gap Excess Ratio (GER) calculations for the topology of Americhelydian lineages recovered in the weighted parsimony analysis. The fossil occurrence of *Allopleuron hofmanni* is used as the maximum age constraint for Pan-Dermochelyidae. (c) GER calculations for the topology of Americhelydian lineages recovered in most previous analyses of fossil turtles. The fossil occurrence of *Eosphargis breineri* is used as the maximum age constraint for Pan-Dermochelyidae. In both GER calculations (b,c), the fossil occurrence of the oldest unambiguous pan-cheloniid, *Euclastes weilandi*, is used as the maximum age constraint for Pan-Cheloniidae. MIG = minimum implied gap; Gmax = maximum gap; Gmin = minimum gap. See electronic supplementary material for sources of fossil occurrence data.

170x137mm (300 x 300 DPI)

Time calibrated phylogeny of Pan-Chelonioida with biogeographic occurrences of each species and a geochemical proxy for palaeoclimatic shifts (O isotopes [50]). AABI = Aptian/Albian boundary interval; KTM = Cretaceous thermal maximum; PETM = Paleocene/Eocene thermal maximum; MECO = Middle Eocene climatic optimum. See electronic supplementary material for sources of fossil occurrence data.

160x152mm (300 x 300 DPI)

Appendix B

Dear Dr. Andrew Dunn,

We thank you for sending us reviews and for your time in handling our manuscript. Reviewer's constructive and helpful comments greatly strengthen and improve our study. I attached replies to each comment from the reviewers. Our general changes are the addition of detailed figures and descriptions, statement about phylogeny on neosuchians, and ICZN code acquisition. Please let us know if any further changes are needed. We are looking forward to hearing from you soon and we hope to publish our manuscript in Royal Society Open Science.

Sincerely,

Junki Yoshida, Ph.D.

Hokkaido University

Sapporo, Japan

junkiyoshida9@gmail.com

Reply to reviewer 1

Great work on a wonderful specimen.

Thank you so much.

Reply to reviewer 2

Overall, I suggest including a larger figure of the skull of the specimen. The material certainly deserves larger resolution photos and page restrictions no longer apply.

As requested, we add a close-up view of the palatine-pterygoid region to provide details and our interpretation of the structure.

Similarly, the authors could illustrate some of their observations derived from the dissecting besides the schematic representation.

Indeed. We add pictures of the dissected hyolingual musculoskeletal elements in the revised manuscript.

I disagree with some details of the conclusions: given the instability of thalattosuchia in phylogenies and the poor sampling of ceratobranchials in the tree makes it difficult to trace the origin of the gular valve closing.

Three possible phylogenetic positions of Thalattosuchia have been proposed in other papers (e.g. Godoy et al., 2019), placing Thalattosuchia out of the least inclusive clade of Goniopholididae + Paralligatoridae + Eusuchia. Our phylogenetic results support this result. Therefore, instability of Thalattosuchia does not matter much to our discussion. Also, the sample size in this study is statistically large (figure 6). In addition, as shown in figure S3, the similar scores of ceratobranchials among neosuchians suggest their synapomorphies, rather than convergence.

Moreover, while the hypothesis of the authors regarding the function of the curved ceratobranchial makes perfect sense, it is a hypothesis not a fact (only a single event in the phylogeny) and the evolutionary implications should be proposed with less certainty. Although our explanation is convincing, we change the statement into the suggestive statement, as reviewer 2 pointed.

Reviewer 2 recommended us to rewrite “Sealed off from one another fold of the gular valve of the tongue”.

Modern crocodylians actively raise the ventral fold of the gular valve to seal the narial and oral regions off, not by moving the dorsal fold. To highlight the form and function relationship between the ventral fold and hyoid, we wrote “These regions can be sealed

off from each other by raising the ventral fold of gular valve at the base of a tongue to contact the dorsal fold at the posterior end of the palate.”

They can't breathe underwater. You probably mean that when their mouth is open underwater (due to the prey) they can still breathe as long as their external nostrils are above the surface.

We rewrite “While opening the mouth underwater to hold prey in, the combination of the secondary palate and the closed ventral valve allows crocodylians to breathe as long as their external nostrils are above the surface.”

This (page3, line46) is a bit confusing and would be better to write: ...which already evolved in the stem lineage of Eusuchia such as... Or if it evolved convergently then please specify.

We simply rewrite “In the evolutionary history of crocodyliforms, the pterygoid secondary palate is present in neosuchian crocodyliforms, such as Eusuchia, paralligatorids, and derived goniopholidids.”

You mean the centrum and the neural arch? That would be surprising because otherwise the specimen is large and the cranial elements are fused.

Yes, the neurocentral sutures are unfused in the presacral and sacral vertebra, which surprisingly means that this individual was still on growing. However, the maximum body size of this species is not the scope of this paper.

This sentence (page8, line 257-258) is confusing, use incorrect syntax, and not too relevant in the light of the previous sentence, I suggest omitting.

We rewrite “Therefore, the goniopholidid covered palate could have been common with modern crocodylians, which is similar with the other covered regions (suborbital fenestra and incisive foramen) by soft tissue in the palate.”

Synapomorphic? How can you establish the correlation between ceratobranchial and oral cavity? + note that Thalattosuchia is more basal in most other recent phylogenies.

We measure the mandible width which is a proxy for oral cavity.

Also, the phylogenetic positions of Thalattosuchia proposed by some recent papers are

still in dispute and chaos although our data still supports that Thalattosuchia is a sister clade of Neosuchia. However, regardless of the recent phylogenetic instability of Thalattosuchia, neosuchians already have retained short ceratobranchials. Therefore, we rewrite “In thalattosuchians, the sister group of Neosuchia in this study, the ceratobranchial is short and straight (figure 6a), indicating that oral enlargement have already retained in Neosuchia or more basal clade (Thalattosuchia+Neosuchia).”

Reviewer 2 recommended us to write “partially submerged and while the mouth being open under water”

We rewrite “this taxon and other extinct neosuchians could have raised the ventral fold of the gular valve in a similar manner to modern crocodylians allowing them to breath while opening the mouth underwater when their external nares were above the water surface.”

Why Batrachomimus is excluded from this figure (figure 16)?

As mentioned in our discussion, *Batrachomimus* show dorsal deflection of ceratobranchial as in other neosuchians, but an outlier in the ratio of ceratobranchial length to mandible width among neosuchians, possibly due to its narrow mandible.

Reply to reviewer 3

There are some odd comments here and there, such as this from the introduction:

“In the evolutionary history of crocodyliforms, the presence of the secondary palate is a key character for eusuchian crocodyliforms, and it appeared in the successive stem groups of Eusuchia such as paralligatorids and derived goniopholidids “

All mesoeucrocodylians have a maxillopalatine secondary palate, and there is an independent maxillopalatine secondary palate in Shartegosuchoidea (even an independent acquisition of the ‘pterygoid secondary palate’). I’m not entirely sure what the authors are meaning here.

We modify the statement into “In the evolutionary history of crocodyliforms, the pterygoid secondary palate is appeared in neosuchian crocodyliforms, such as Eusuchia, paralligatorids, and derived goniopholidids.”

Terminology of “secondary palate” is obscure in many of the previous papers, and as Reviewer 3 indicated, we use “pterygoid secondary palatine” for clarification.

“Goniopholidids, which is a transitional group to Eusuchia,” Goniopholidids aren’t. It’s an odd thing to say. And there are more of these. I think the authors need to give a serious re-think to the tone of the manuscript. It reads very teleological, with neosuchian groups being ‘pit-stops’ on the road to Eusuchia. They weren’t, many, such as goniopholidids, were evolutionary successful lineages that persisted for tens of millions of years.

...As the authors will no doubt be aware crocodylomorph phylogeny is chaotic in places. The position of major clades are in flux. So the comment: “Goniopholididae is a group of basal neosuchian crocodyliforms found in the in the northern hemisphere during the Jurassic and Early Cretaceous, and a sister clade to Paralligatoridae and Eusuchia” Needs to be better cited. I would simply state that Goniopholididae is a successful lineage of neosuchians outside of the Paralligatoridae + Eusuchia clade. That way, what you are saying doesn’t require discussing competing phylogenetic hypotheses.

Considering the phylogenetic dispute within neosuchians, we change the statement as reviewer 3 suggested. Thank you.

Also, there is a completely different topology used for the R analyses in the SI... Where

does it come from? Why are you not using the topology you analysed with the modified Andrade dataset?

For visualization, we used heated trees in figures. Non-scored terminal taxa are pruned from our obtained tree, and then the topology is simplified. A tree in figure 16 comes from figure S1-3, and trees from figure S1-3 comes from our obtained tree (figure 13).

As RSOS is an online-only journal can the authors make sure everything they do is done in accordance with the ICZN Code? That means ensuring at proof stage the ZooBank LSIDs for both the paper and the new species are present.

We are ready now. Thank you.

So, the anteriorly elongated internal nares. I can't see it in the figures. There are no close-ups, only interpretative line drawings. And from those, I would hesitate to say the morphology isn't purely post-mortem damage. To be honest, the description and figuring are what frustrate me most. As the authors say, this is the most complete known skeleton of a goniopholidid, and the description is largely in the SI and so are the figures. The complete description needs to be 'main text', there needs to be much more, and better figures.

As reviewer 3 requested, we add the close-up view of internal choana and its surrounding region as figure 2 and put all specimen figures and description into the main text.

Appendix C

Dear Dr. Andrew Dunn,

We thank you for sending us reviews and for your time in handling our manuscript. Reviewer's constructive and helpful comments greatly improve our study. I attached replies to each comment from the reviewers. As suggested, we modified English words and phrases in our revised manuscript. We are looking forward to hearing from you soon and we hope to publish our manuscript in Royal Society Open Science.

Sincerely,

Junki Yoshida, Ph.D.

Hokkaido University Museum

Sapporo, Japan

junkiyoshida9@gmail.com

Reply to reviewer 2

I suggest specifying that the instable phylogenetic position of Thalattosuchia is not affecting the conclusions (if that is the case). think the manuscript is otherwise acceptable once minor improvements of the English are made here and there.

Thank you. We added a sentence and modified some parts of English writing.

Reply to reviewer 4

Congratulations, your work is very interesting and valuable for crocodyliforms anatomy, especially the evolution from neosuchia to modern crocodylians.

Thank you so much.